# Protein arginine methylation facilitates KCNQ channel-PIP$_2$ interaction leading to seizure suppression

Hyun-Ji Kim[1†], Myong-Ho Jeong[2†], Kyung-Ran Kim[3,4†], Chang-Yun Jung[2†], Seul-Yi Lee[1], Hanna Kim[1], Jewoo Koh[1], Tuan Anh Vuong[2], Seungmoon Jung[5], Hyunwoo Yang[5], Su-Kyung Park[2], Dahee Choi[2,6], Sung Hun Kim[7], KyeongJin Kang[8], Jong-Woo Sohn[9], Joo Min Park[10], Daejong Jeon[11,12], Seung-Hoi Koo[6], Won-Kyung Ho[3,4], Jong-Sun Kang[2*], Seong-Tae Kim[2*], Hana Cho[1*]

[1]Department of Physiology, Samsung Biomedical Institute, Sungkyunkwan University School of Medicine, Suwon, Korea; [2]Department of Molecular Cell Biology, Samsung Biomedical Institute, Sungkyunkwan University School of Medicine, Suwon, Korea; [3]Department of Physiology and bioMembrane Plasticity Research Center, Seoul National University College of Medicine, Seoul, Korea; [4]Neuroscience Research Institute, Seoul National University Medical Research Center, Seoul, Korea; [5]Department of Bio and Brain Engineering, Korea Advanced Institute of Science and Technology, Daejeon, Korea; [6]Division of Life Sciences, Korea University, Seoul, Korea; [7]Department of Neurology, College of Medicine, Kangwon National University, Chuncheon, Korea; [8]Department of Anatomy and Cell Biology, Sungkyunkwan University School of Medicine, Suwon, Korea; [9]Department of Biological Sciences, Korea Advanced Institute of Science and Technology, Daejeon, Korea; [10]Center for Cognition and Sociality, Institute for Basic Science, Daejeon, Korea; [11]Department of Neurology, Laboratory for Neurotherapeutics, Comprehensive Epilepsy Center, Seoul National University Hospital, Seoul, Korea; [12]Advanced Neural Technologies, Seoul, Republic of Korea

*For correspondence: kangj01@skku.edu (J-SK); stkim@skku.edu (S-TK); hanacho@skku.edu (HC)

†These authors contributed equally to this work

Competing interests: The authors declare that no competing interests exist.

**Abstract** KCNQ channels are critical determinants of neuronal excitability, thus emerging as a novel target of anti-epileptic drugs. To date, the mechanisms of KCNQ channel modulation have been mostly characterized to be inhibitory via Gq-coupled receptors, Ca$^{2+}$/CaM, and protein kinase C. Here we demonstrate that methylation of KCNQ by protein arginine methyltransferase 1 (Prmt1) positively regulates KCNQ channel activity, thereby preventing neuronal hyperexcitability. *Prmt1 +/-* mice exhibit epileptic seizures. Methylation of KCNQ2 channels at 4 arginine residues by Prmt1 enhances PIP$_2$ binding, and *Prmt1* depletion lowers PIP$_2$ affinity of KCNQ2 channels and thereby the channel activities. Consistently, exogenous PIP$_2$ addition to *Prmt1+/-* neurons restores KCNQ currents and neuronal excitability to the WT level. Collectively, we propose that Prmt1-dependent facilitation of KCNQ-PIP$_2$ interaction underlies the positive regulation of KCNQ activity by arginine methylation, which may serve as a key target for prevention of neuronal hyperexcitability and seizures.

**eLife digest** In the brain, cells called neurons transmit information along their length in the form of electrical signals. To generate electrical signals, ions move into and out of neurons through ion channel proteins – such as the KCNQ channel – in the surface of these cells, which open and close to control the electrical response of the neuron.

Abnormally intense bursts of electrical activity from many neurons at once can cause seizures such as those experienced by people with epilepsy. A significant proportion of patients do not respond to current anti-seizure medications. Openers of KCNQ channels have emerged as a potential new class of anti-epileptic drugs. A better understanding of how KCNQ channels work, and how their opening by $PIP_2$ lipid signals is regulated, could help to develop more effective therapies for epilepsy.

A process called methylation controls many biological tasks by changing the structure of key proteins inside cells. Although methylation occurs throughout the brain, its role in controlling how easily neurons are activated (a property known as "excitability") remains unclear.

Kim, Jeong, Kim, Jung et al. now show that a protein called Prmt1 methylates the KCNQ channels in mice, and that this methylation is essential for suppressing seizures. Mice born without the Prmt1 protein developed epileptic seizures and the KCNQ channels in their neurons featured a reduced level of methylation. However, increasing the amount of $PIP_2$ in these neurons restored their excitability back to normal levels. The methylation of KCNQ channel proteins increases their affinity for PIP2, which is critical to open KCNQ channels. Kim et al. propose that these "opening" controllers balance the action of known "closers" of KCNQ channels to maintain neurons in a healthy condition.

In future, Kim et al. plan to investigate whether methylation affects the activity of other ion channels controlled by $PIP_2$. Such experiments will complement a more widespread investigation into other ways in which the Prtmt1 protein may control the activity of neurons.

## Introduction

Epilepsy imposes a major burden at both global and individual levels. Worldwide, about 1% of the population suffers from epilepsy, and nearly 4% of the population will experience epilepsy at some point during their lifetime (*Malkki, 2014*). In almost 30% of patients with epilepsy, anti-seizure medications do not provide sufficient seizure control (*Malkki, 2014*). Thus, understanding the etiology of epilepsy is essential both for clinical management of patients and for conducting neurobiological research that will direct future therapies. The etiology of epilepsy was formerly regarded as unknown in about three-quarters of patients; however, massively parallel gene-sequencing studies showed the importance of gene mutations in the etiology of epilepsy (*Thomas and Berkovic, 2014*). Among those epileptic conditions linked to channelopathies, mutations in potassium channel subunits represent the largest category (*Brenner and Wilcox, 2012*; *Cooper, 2012*; *Noebels, 2003*). Together with KCNA1, KCNQ2 and KCNQ3 are the earliest identified K$^+$ channels associated with idiopathic epilepsy (*Rogawski, 2000*; *Schroeder et al., 1998*; *Browne et al., 1994*). More than 30 mutations have been detected in KCNQ channels, most of which were found in KCNQ2: only four mutations were found in KCNQ3 (*Maljevic et al., 2010*). These mutations caused a reduction in channel activity by different molecular mechanisms, by which all can lead to membrane depolarization and increased neuronal firing.

In the central nervous system (CNS), heteromeric KCNQ2/3 potassium channels form the M-current, a subthreshold potassium current (*Delmas and Brown, 2005*; *Jentsch, 2000*). Due to the slow gating kinetics, M-currents effectively oppose sustained membrane depolarization and repetitive action potential (AP) firing (*Brown and Passmore, 2009*). Thus, breakdown of M-channels by loss of function mutations or pharmacological inhibitors leads to neuronal hyperexcitability (*Schroeder et al., 1998*; *Delmas and Brown, 2005*). KCNQ potassium channels are standouts for epileptologists, in that they are both mutated in human epilepsy and principal targets of an approved anti-epileptic drug (ezogabine/retigabine) (*Gunthorpe et al., 2012*; *Soldovieri et al., 2011*). Therefore, regulation of KCNQ channel activity has been studied by many groups

(*Delmas and Brown, 2005*). Notably, M-channels are inhibited by muscarinic acetylcholine receptor agonists, leading to a profound increase in cellular excitability that can be reversed by the withdrawal of receptor agonist. Some subtypes of receptors for dopamine, serotonin, glutamate, and several peptide neurotransmitters, including luteinizing hormone-releasing hormone and bradykinin, are also found to be capable of inhibiting the M-channels. An important property of M-channels is that they require membrane phosphatidylinositol-4,5-bisphosphate ($PIP_2$) to open (*Delmas and Brown, 2005*; *Suh and Hille, 2008*), and the certain receptors mentioned above suppress M-current by depletion of $PIP_2$ (*Delmas and Brown, 2005*). In addition, M-channels are also inhibited by CaM in a $Ca^{2+}$-dependent manner, providing another mode of intracellular modulation of M-channel activities (*Gamper and Shapiro, 2003*). Recent work indicates that protein kinase C (PKC) phosphorylation of the KCNQ2 subunit reduces the affinity for its $PIP_2$ binding and activities (*Kosenko et al., 2012*; *Lee et al., 2010*). Likewise, sumoylation of KCNQ channels results in reduced channel activity (*Qi et al., 2014*). Thus, all regulatory pathways discovered so far act on the KCNQ channels to reduce the channel activity and increase neuronal excitability. Given its important role in stabilization of neuronal excitability, *there has* been much speculation for regulatory mechanisms to increase the channel-$PIP_2$ interaction and channel activities, however, to date *there is no* evidence supporting such regulatory mechanism.

Protein methylation, along with phosphorylation, controls a variety of cellular functions (*Nicholson et al., 2009*). Protein arginine methyltransferases (Prmts) are enzymes that catalyze the transfer of a methyl group to arginine residues of histone or non-histone substrates (*Boisvert et al., 2005*). In mammals, nine Prmts have been characterized. Among these, Prmt1, originally identified as a histone H4 methyltransferase, methylates many non-histone proteins and implicated in diverse cellular processes including RNA processing, transcriptional regulation, oncogenesis, cell survival, insulin signaling, and metabolism (*Boisvert et al., 2005*; *Bedford and Clarke, 2009*; *Krause et al., 2007*). Although Prmt1 is a predominant Prmt in mammalian cells and is highly expressed in the CNS (*Nicholson et al., 2009*; *Bedford and Clarke, 2009*), its functional significance in the CNS has not yet been identified.

The positively charged (basic) arginines or lysines are candidates for mediating electrostatic interaction with $PIP_2$ in channels such as KCNQ (*Hernandez et al., 2008*), Kir2 (*Hansen et al., 2011*; *Huang et al., 1998*; *Lopes et al., 2002*), and GIRK (*Whorton and MacKinnon, 2011*). Considering that each additional methyl group to an arginine residue can readily modulate their physical properties (*Bedford and Clarke, 2009*), methylation of arginine residues in $PIP_2$ binding domain may alter KCNQ channels' affinity for $PIP_2$. However it is not known whether such methylation really occurs and regulates the channel activity and whether it is implicated in common disease phenotypes. In the present study, the role of arginine methylation of KCNQ in regulation of channel activities and neuronal excitability was investigated. *Prmt1*-heterozygous (+/-) mice exhibit epileptic seizures. We also found that *Prmt1* depletion causes a decreased interaction between $PIP_2$ and KCNQ channels, consequently causing a reduction in KCNQ channel activity. Prmt1 interacts and methylates at 4 arginine residues of KCNQ channels. Hippocampal neurons from the heterozygote mice lack KCNQ currents, and the current can be restored by exogenous $PIP_2$ addition, accompanied by concomitant rescue of normal excitability. Furthermore a pharmacological inhibition of methylation or methylation-deficient mutants of KCNQ2 reduce $PIP_2$ binding and activities of KCNQ channels. These data demonstrate that protein arginine methylation facilitates KCNQ channel-$PIP_2$ interaction, leading to seizure suppression. We propose that Prmt1-dependent regulation of KCNQ channels represents an important mechanism of neuronal protection against over-excitability.

## Results

### *Prmt1+/-* mice show spontaneous seizure activity

To assess the physiological importance of Prmt1 in the CNS, we utilized mutant mice for the *Prmt1* gene in a C57BL/6J background (*Choi et al., 2012*). As *Prmt1* homozygous knockout mice are embryonic lethal (*Choi et al., 2012*), we used heterozygous mice (*Prmt1+/-*) for our study. Immunoblot analysis demonstrated that Prmt1 proteins were highly expressed in wild-type (WT) brain but significantly reduced in the brain of *Prmt1+/-* mice (*Figure 1a*). We used long-term video- electroencephalographic (EEG) recording (24 hr per day for 6 days) to monitor behavior and spontaneous

EEG activity in freely moving *Prmt1+/-* (*n* = 8) and WT mice (*n* = 6), which led to observation of spontaneous seizure activity from *Prmt1+/-* mice (*n* = 8) (**Figure 1b and c**). Epileptiform spikes with a delta frequency range (1–3 Hz) appeared in *Prmt1 +/-* mice (**Figure 1d**, **Figure 1—source data 1**), and lasted for 96.7 ± 12.5 s (range, 30–400 s, **Figure 1e**, **Figure 1—source data 1**). These results are in parallel with human seizures that usually last less than 3 min (**Bromfield EB and Sirven, 2006**). The number of seizure occurrences was 4.1 ± 1.4 per day in *Prmt1+/-* mice (range, 0.3–12 per day, **Figure 1f**, **Figure 1—source data 1**). Seizure activity was not observed in WT mice (**Figure 1d–f**,

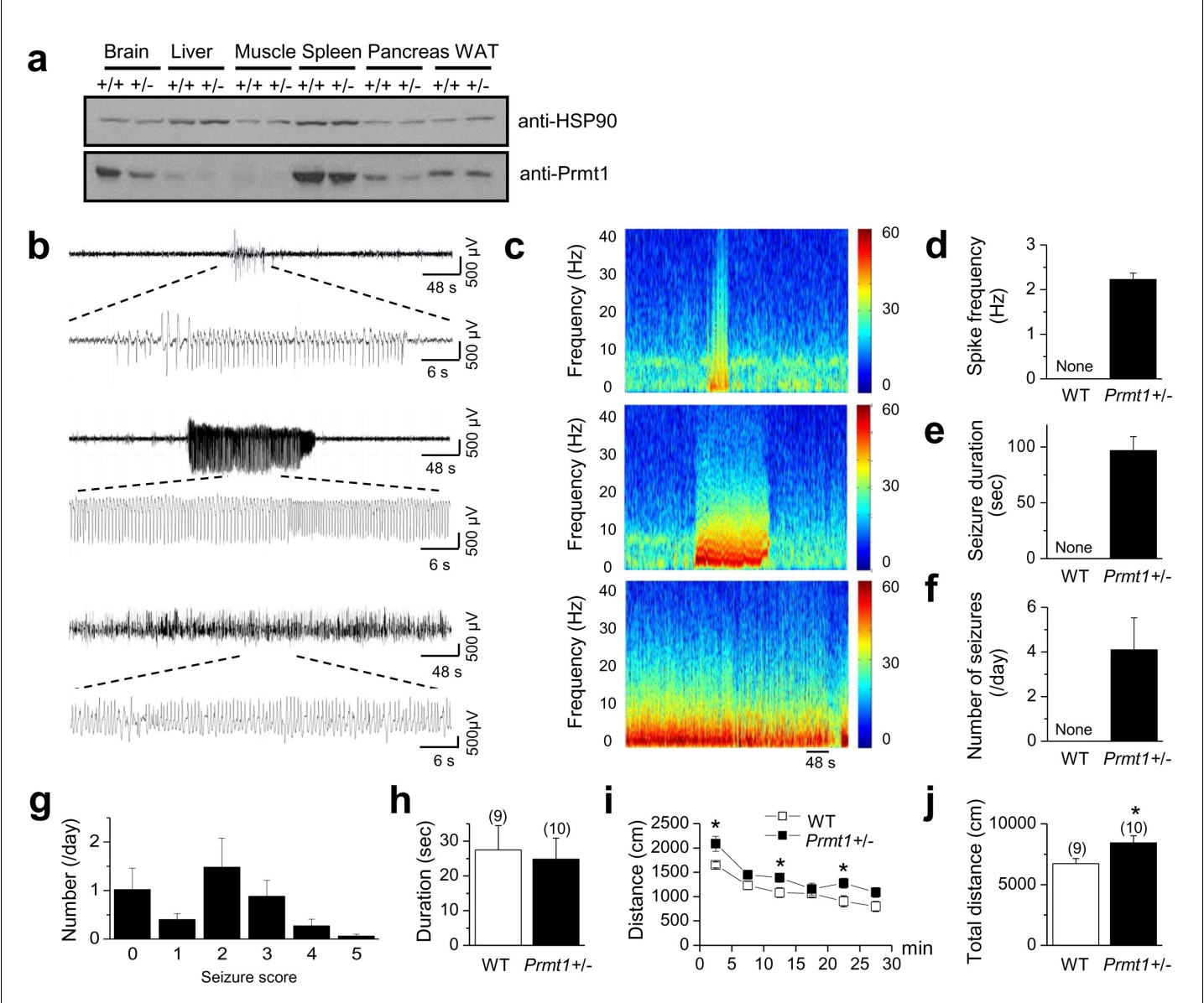

**Figure 1.** Spontaneous seizures and increased locomotor activities in *Prmt1+/-* mice. (**a**) reduced expression of Prmt1 in *Prmt1+/-* mice, compared to WT control mice. (**b**) Representative traces of seizure activities: short (upper) or long (middle and lower) duration of epileptiform activity. (**c**) A colored power spectrum of the trace shown in **b**. (**d–g**) The mean bar graphs of seizure spike frequency (**d**) duration (**e**), number of seizures per day (**f**), and seizure scores (**g**) in WT (*n* = 6) and *Prmt1+/-* mice (*n* = 8). (**h**) The WT and the *Prmt1+/-* mice spent similar amounts of times in the center of the open-field box. (**i**) The *Prmt1+/-* mice moved a longer distance than did the WT mice. (**j**) Total distance moved for 30 min. *p<0.05 by Student's *t*-test.

The following source data is available for figure 1:

**Source data 1.** Source data for **Figure 1**.

Figure 1—source data 1). During these seizure activities of *Prmt1+/-* mice, we typically observed partial seizure behaviors (stages 1–3 seizures on the Racine scale) or no changes in behavior (*Figure 1g*, *Figure 1—source data 1*). Convulsive seizures (stages 4 and 5) were rarely observed in *Prmt1+/-* mice (*Figure 1g*).

Behavioral hyperactivity, such as an increase in open-field locomotion, frequently accompanies seizures (*Kim et al., 2011*; *Peñagarikano et al., 2011*; *Peters et al., 2005*). In an open-field test, male mutants showed a slight increase in locomotor activity (*Figure 1h–j*, *Figure 1—source data 1*). The *Prmt1+/-* mice (*n* = 10) spent a similar amount of time in the center of the open-field box compared with the control mice (*n* = 9) (*Figure 1h*). However, the *Prmt1+/-* mice moved a longer distance than WT mice (*Figure 1i*). The total travel distance was increased significantly in *Prmt1+/-* mice, compared to that in WT mice ($p < 0.05$; *Figure 1j*). Thus, *Prmt1+/-* mice display signs of a persistent neuronal hyperexcitability, including spontaneous seizures and a slightly increased locomotor activity.

## Enhanced excitability and input resistance of dentate granule cells in *Prmt1+/-* mice

Hyperexcitability of hippocampal neurons is a characteristic feature of most epilepsies (*Noebels, 2003*; *McCormick and Contreras, 2001*). To identify neural mechanisms underlying spontaneous seizure activities in *Prmt1+/-* mice, we performed electrophysiological recordings from the dentate gyrus granule cells (GCs) of hippocampal slices from WT and mutant mice. GCs from WT mice typically displayed tonic firing patterns in response to a 1-s square current pulse injection: AP frequency elevated as the magnitude of the square pulse increased (*Figure 2a and c*, *Figure 2—source data 1*). GCs from *Prmt1+/-* mice showed significantly higher AP frequency than those from WT mice (*Figure 2b and c*, *Figure 2—source data 1*). The averaged AP frequency in response to a 200 pA depolarizing current in WT GCs was $8.1 \pm 1.4$ Hz (*n* = 16), while it increased significantly to $31.2 \pm 2.2$ Hz (*n* = 18, $p < 0.01$) in *Prmt1+/-* GCs (*Figure 2c*). These data indicate that the heterozygous deletion of the *Prmt1* gene enhanced excitability of hippocampal neurons. To determine whether the increased firing resulted from a change in the threshold current, we compared the magnitude of the current injection to reach a threshold for AP firing (AP threshold current) in WT and *Prmt1+/-* GCs. Representative traces in *Figure 2d,e* clearly show that the current for spike initiation was reduced in *Prmt1+/-* neurons. Analysis of pooled results revealed a significant decrease in the mean threshold current in *Prmt1+/-* GCs compared to that in WT GCs (*Figure 2f*, *Figure 2—source data 1*). As illustrated in *Figure 2g*, the change in threshold current was correlated with an increase in the input resistance. The amplitude and duration of APs obtained in WT were not altered by the reduction of the Prmt1 gene dose (*Figure 2h and i*, *Figure 2—source data 1*). Therefore, the change in the threshold current in *Prmt1+/-* neurons reflects the effect on input resistance, which facilitates their AP initiation and, hence, increase in spike number.

## Reduced KCNQ channel activity causes the neuronal hyperexcitability in *Prmt1+/- mice*

Next, we explored the ionic mechanisms underlying the increased excitability. An increase in neuronal excitability without a change in AP height and width indicated that the kinetics and amplitude of voltage-gated currents associated with the spike itself were not affected. Increased cation current or leak current was unlikely to be involved in the phenotype, either, as this would be expected to result in a decrease in input resistance. Thus, the most probable target was the KCNQ/M channel. KCNQs are voltage-dependent $K^+$ channels that partially open at the resting membrane potential. Inhibiting the KCNQ channel increases both input resistance and neuronal excitability (*Jentsch, 2000*; *Brown and Passmore, 2009*). Thus, we examined whether M-channel deficiency contributed to increased neuronal excitability of *Prmt1+/-* mice. The M-current amplitude was measured using a standard deactivation voltage protocol (*Peters et al., 2005*; *Lawrence et al., 2006*; *Adams et al., 1982*). In GCs from WT mice, a step to -60 mV forced deactivation of a slowly relaxing outward current that decayed, consistent with the presence of M-current. The M-channel antagonist XE991 (10 μM) caused a complete loss of detectable M-current (*Figure 3a*), accompanied by a reduction in the holding current. However, *Prmt1+/-* GCs showed little measurable M-currents ($p < 0.001$) (*Figure 3b–c*, *Figure 3—source data 1*).

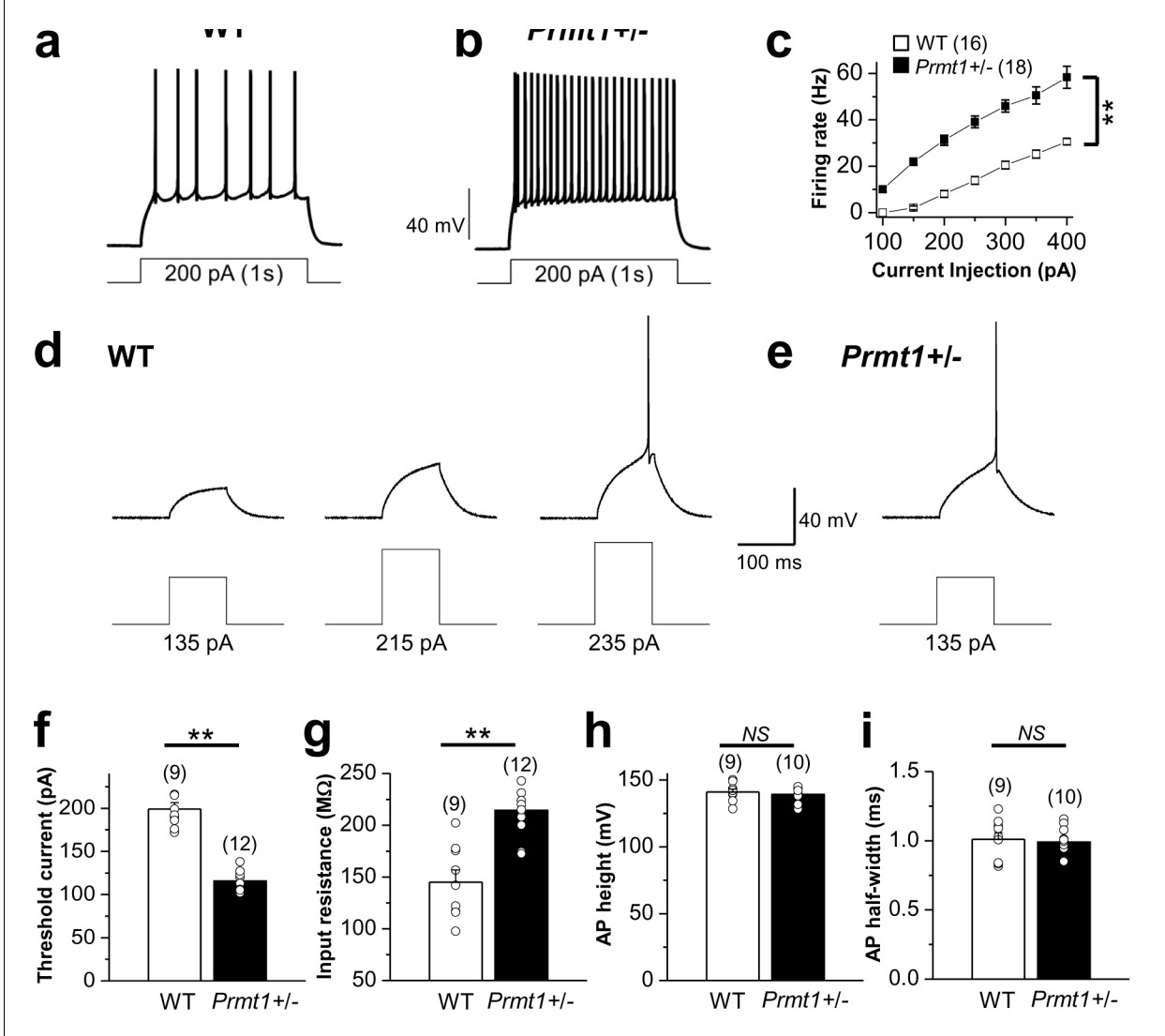

**Figure 2.** Comparison of neuronal excitability in WT and *Prmt1+/-* mice. (a–b) a and b panels show representative trace in the whole-cell current-clamp recording from WT (a) and *Prmt1+/-* (b) mature dentate GCs in response to 1-s depolarizing current injection (200 pA), respectively. (c) the mean number of action potentials (AP No.) plotted against the eliciting currents (from 100 pA to 400 pA, + 50 pA increment, during 1-s). At all amplitudes, the Mean ± S.E.M. AP No. is significantly higher in *Prmt1+/-* (■; n = 18, seven mice) than WT dentate GCs (□; n = 16, seven mice; p<0.01). (d–e) the threshold current for single AP elicited by a short depolarizing (100 ms) step pulse of various amplitude in WT (d) and *Prmt1+/-* (e) dentate GCs. (f–i) the mean value of threshold current for AP generation (100 ms duration; f), input resistance (g), AP height (h), and AP half-width (i) from WT and *Prmt1 +/-* mature dentate GCs.

The following source data is available for figure 2:

**Source data 1.** Source data for *Figure 2*.

Consistent with the major contribution of the KCNQ channel (*Delmas and Brown, 2005*), XE991 application led to a depolarization (ΔVm, 4.5 ± 1.6 mV) in WT neurons. In contrast, such an effect of XE991 was abolished in the mutant (ΔVm, −0.9 ± 0.7 mV) (*Figure 3—figure supplement 1*, *Figure 3—figure supplement 1—source data 1*). Similarly, application of XE991 increased input resistance in the WT neurons (Δinput resistance, 51.7 ± 4.6 MΩ) but had less of an effect in the mutant (Δinput resistance, 29.2 ± 6.3 MΩ) (*Figure 3d*, *Figure 3—source data 1*). The *Prmt1+/-* neurons' threshold currents were also insensitive to XE991 treatment (*Figure 3e*, *Figure 3—source data 1*).

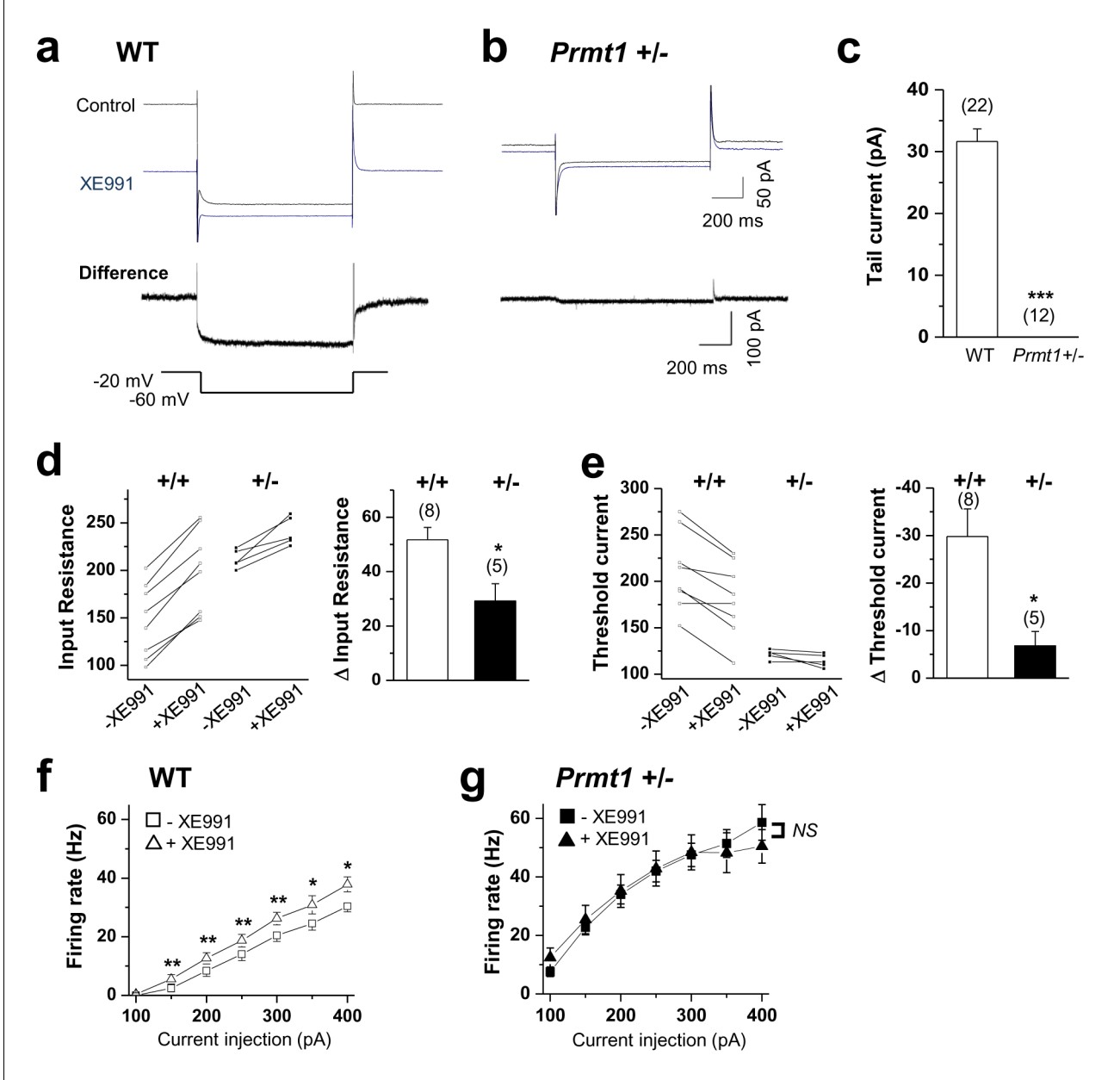

**Figure 3.** KCNQ current deficiency contributed to the persistent hyperexcitability in *Prmt1*+/- mice. (a–b) Representative current traces of voltage clamp recordings from WT (a; *n = 22, seven mice*) and *Prmt1*+/- (b; *n = 12, four mice*) mature dentate GCs in response to the voltage protocol depicted below. The upper panel shows the overlay of currents in the absence (black) or presence (blue) of 10 µM XE991. To illustrate the kinetic components of M-current relaxation and activation, the corresponding difference current is shown in an expanded scale below each current trace. (c) Summary statistics from experiments shown in a–b reveal loss of M-current in *Prmt1*+/- mice. Error bars, S.E.M. ***p<0.001 by Student's *t*-test. (d–e) Changes in input resistance (d) and threshold current (e) in WT (+/+) and *Prmt1*+/- GCs (+/-) in response to 10 µM XE991. Each connected line represents an individual neuron. Right panel displays the summary of changes in input resistance or threshold current upon application of 10 µM XE991. XE991 was less effective in mutants than in WT. (f–g) Average numbers of APs generated during incremental 1-s depolarizing current steps before and after application of XE991 in WT (f) and *Prmt1*+/- GCs (g). XE991 produced a substantial change in firing rate of WT GCs (f; *n = 7~11, seven mice*), *whereas* little change in firing rate occurred with XE991 in *Prmt1*+/- GCs (g; *n = 9, seven mice*). *p<0.05; **p<0.01 by paired Student's *t*-test.

The following source data and figure supplements are available for figure 3:

**Source data 1.** Source data for *Figure 3*.

**Figure supplement 1.** Changes in membrane potential in WT (+/+) and *Prmt1*+/- GCs (+/-) in response to 10 µM XE991.
*Figure 3 continued on next page*

*Figure 3 continued*

**Figure supplement 1—source data 1.** Source data for *Figure 3—figure supplement 1*.

These results suggest that the increase in the input resistance of *Prmt1*+/- neurons might be a result of the reduced level of KCNQ channel activity.

We further examined the effect of XE991 on firing frequency. The firing frequency in WT neurons increased significantly after applying 10 µM XE991 (*Delmas and Brown, 2005*) (*Figure 3f*, *Figure 3—source data 1*). For example, the AP frequency in response to a 200 pA-depolarizing current was 8.4 ± 1.9 Hz in the control and increased to 12.6 ± 1.9 Hz following the application of 10 µM XE991. This XE991-induced spiking was largely absent in the mutant neurons (*Figure 3g*, *Figure 3—source data 1*). Before and after the XE991 treatment, AP frequency in response to a 200 pA depolarizing current was 34 ± 3.2 Hz and 35.2 ± 5.6 Hz, respectively. Thus, these results showed that *Prmt1*+/- GCs displayed a high firing rate at baseline, and their firings did not further increase during XE-991 application, suggesting that defective M-current contributes to the neuronal hyperexcitability observed in the *Prmt1*+/- mice. However, we cannot exclude the potential involvement of other Prmt1 target(s) in the neuronal hyperexcitability observed in the *Prmt1*+/- mice.

To assess the effect of protein methylation on M-currents, WT hippocampal slices were treated with a pan-methyltransferase inhibitor 5-deoxy-5-(methylthio) adenosine (MTA) or a *m*ore specific blocker of Prmt1, furamidine dihydrochloride. The treatment of WT hippocampal slices with MTA (100 µM) or furamidine dihydrochloride (20 µM) for 1 hr completely abolished M-currents (*Figure 4a–c*; p<0.001 vs WT control, *Figure 4—source data 1*). We then studied their effects on neuronal excitability. The application of MTA (100 µM, 1 hr) enhanced neuronal excitability of WT GCs (*Figure 4d–e*). After the MTA treatment, AP frequency in response to a 200 pA depolarizing current was 34.7 ± 4.6 Hz. Consistent with ablation of M-currents by MTA, subsequent application of XE991 had no further effect (AP frequency was 31.8 ± 3.9 Hz) (*Figure 4d–e*, *Figure 4—source data 1*). The MTA-induced increase in excitability was accompanied by the declined AP threshold currents and increased input resistance (threshold current, 110.5 ± 5.5 pA, *n* = 4; input resistance, 234.0 ± 5.8 MΩ, *n* = 4), which were relatively insensitive to a subsequential treatment with XE991 (p>0.05; *Figure 4j–k*, *Figure 4—source data 1*). Furthermore, the treatment of WT hippocampal slices with furamidine dihydrochloride exhibited a *similar* effect on neuronal excitability, input resistance, and threshold current, confirming the results obtained with MTA treatment (*Figure 4f–g and j–k*, *Figure 4—source data 1*). However, methylation suppression with MTA had no effect on AP firing (*Figure 4h–i*, *Figure 4—source data 1*), threshold current (*Figure 4j*, *Figure 4—source data 1*), or input resistance (*Figure 4k*, *Figure 4—source data 1*) in *Prmt1*+/- GCs. Thus, these data suggest that the reduction of Prmt1 activity in the hippocampal neurons might be responsible for the neuronal hyperexcitability in *Prmt1*+/- neurons.

## Prmt1 interacts with and methylates KCNQ2

Importantly, KCNQ2 and KCNQ3 protein levels were unchanged in hippocampi from *Prmt1*+/- mice, suggesting the involvement of post-translational mechanisms in the regulation of KCNQ currents (*Figure 5—figure supplement 1*). Previous studies have suggested that methylation of some ion channel proteins could affect their activities/functions (*Beltran-Alvarez et al., 2013*; *Sariban-Sohraby et al., 1984*). To investigate a potential interaction between KCNQ2 and Prmt1, co-immunoprecipitation experiments were conducted with HEK293T cells. KCNQ2 and Prmt1 proteins were coprecipitated when co-expressed in HEK293T cells (*Figure 5a*). The intracellular C-terminal region of KCNQ2 encompassing amino acids 320–840, designated as KCNQ2-C, was sufficient to interact with Prmt1 (*Figure 5b and c*). To further confirm the interaction between KCNQ2 and Prmt1 proteins in the native neuronal environment, co-immunoprecipitation was performed with mouse hippocampal lysates. We found that KCNQ2 was co-immunoprecipitated endogenously with Prmt1 in hippocampus (*Figure 5d*).

To assess whether KCNQ2 is methylated in vivo and to identify the in vivo methylation sites, we performed a liquid-chromatography-coupled tandem mass spectrometric analysis. Four arginine

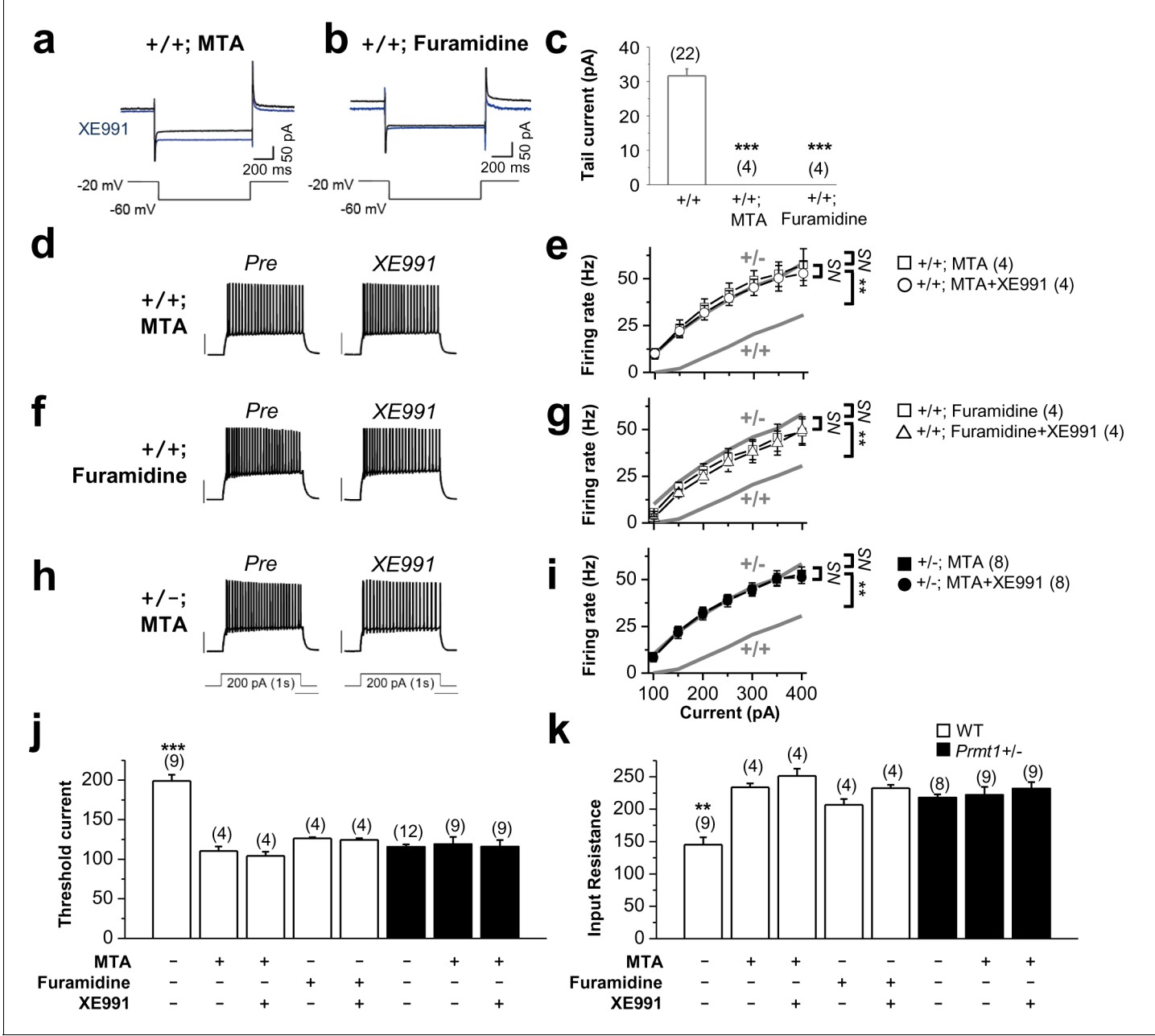

**Figure 4.** Methylation suppression with MTA or furamidine, increases neuronal excitability via KCNQ channel. (a–b) Representative current traces recorded from MTA- (a) or furamidine- (b) pretreated WT GCs using the voltage protocol depicted below in the absence (black) or presence (blue) of 10 µM XE991. (c) Summary of M-currents from experiments shown in a–b. Data for M-currents of WT control shown in *Figure 3c* is also shown as a reference. ***p<0.001 by Student's *t*-test. (d–i) APs were evoked by applying 1-s depolarizing current pulses of different intensities (100–400 pA) in WT (d–g) or *Prmt1*+/- (h–i) neurons. P*anel d, f, and h illustrate* representative traces after *incubation* for 1 hr with 100 µM MTA (d and h) or 20 µM furamidine (f). The effect of applying XE991 once the maximal effect of MTA or furamidine is achieved was shown in right. Scale bar indicate 40 mV. Summarized data compare the number of APs before and after application of XE991 in MTA-pretreated WT (e), furamidine-pretreated WT (g), and MTA-pretreated mutants (i). The gray lines shows the number of APs observed in untreated WT and *Prmt1*+/- neurons. Genotypes are given on each line. Firing rate in MTA-treated WT cells (□; n = 4, two mice) or furamidine-treated WT cells (□; n = 4, three mice) increased to the level of *Prmt1*+/- cells (gray line). Further, XE991 had no effect in MTA-treated WT cells (○; n = 4, paired Student's *t* tests) or furamidine-treated WT cells (△; n = 4, paired Student's *t* tests). MTA had no further effects on *Prmt1*+/- neurons. *NS*, not significantly different. (j–k) the mean value of threshold current for AP generation (j) and input resistance (k). ANOVA Tukey test. **p<0.02; ***p<0.001.

The following source data is available for figure 4:

**Source data 1.** Source data for *Figure 4*.

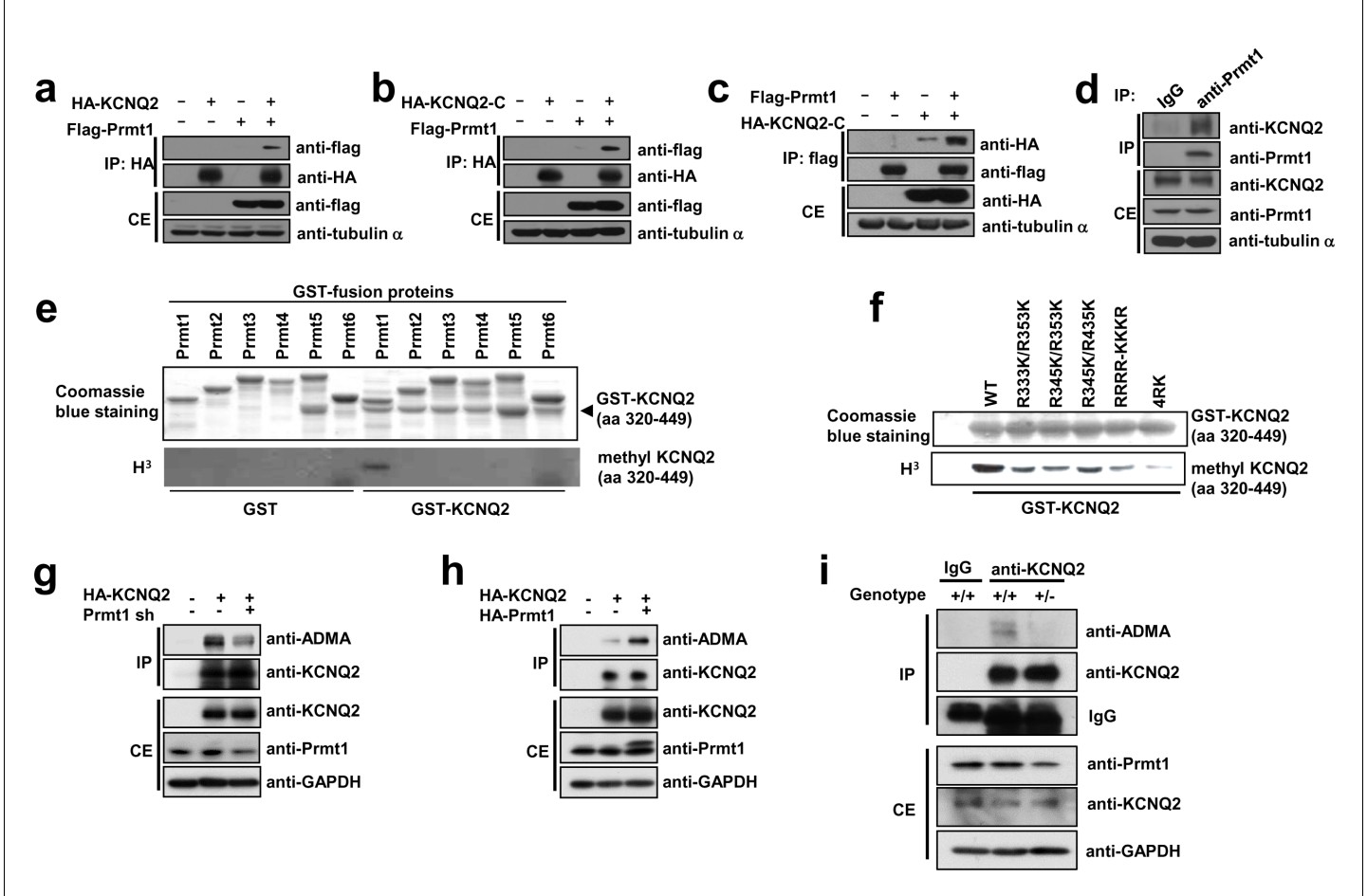

**Figure 5.** Prmt1 binds to and methylates KCNQ2 at R333, R345, R353, and R435. (**a–c**) Immunoblotting analysis showing the physical association of KCNQ2 and Prmt1. HEK293T cells were transfected with expression vectors, as indicated. Whole-cell lysates were immunoprecipitated and immunoblotted by either anti-flag or anti-HA antibody. Representative data from at least three independent experiments are shown. (**d**) Western blotting analysis showing endogenous interaction of KCNQ2 and Prmt1. Cell lysates from mouse hippocampus were immunoprecipitated with anti-Prmt1 antibody and were immunoblotted with anti-KCNQ2 antibody or anti-Prmt1 antibody. Representative data from at least three independent experiments are shown. (**e**) In vitro methylation assays with GST-KCNQ2 (amino acids 320–449) and a series of GST-Prmts (1–6) in the presence of [$^3$H]S-adenosylmethionine (SAM). Total amounts of GST-KCNQ2 (arrowhead) and GST-Prmts are shown by Coomassie brilliant blue staining. (**f**) In vitro methylation assays with GST-Prmt1 together with GST-KCNQ2 (amino acids 320–449) WT, R333K/R353K, R345K/R353K R345K/R435K. R333K/R345K/R353K (RRRR-KKKR), or R333K/R345K/R353K/R435K (4RK) in the presence of [$^3$H]SAM. (**g**) Immunoblotting analysis showing the decreased asymmetric dimethylation of KCNQ2 in *Prmt1* knockdown cells. HEK293T cells were transfected with control or HA-KCNQ2 in combination with control or *Prmt1* shRNA expression vectors, followed by immunoprecipitation with KCNQ2 antibodies and immunoblotting with antibodies to ADMA, KCNQ2, Prmt1 and GAPDH. (**h**) Immunoblotting analysis showing the enhanced asymmetric dimethylation of KCNQ2 in Prmt1 overexpressing cells. HEK293T cells were transfected with control or HA-KCNQ2 in combination with control or Prmt1 expression vectors, followed by immunoprecipitation with KCNQ2 antibodies. (**i**) Immunoblotting analysis showing decreased asymmetric dimethylation of KCNQ2 in *Prmt1*+/- brain, compared to the WT control. Brain lysates from *Prmt1*+/+ and *Prmt1*+/- mice were immunoprecipitated with anti-KCNQ2 antibodies or control IgG.

The following figure supplements are available for figure 5:

**Figure supplement 1.** Expression of KCNQ2 and KCNQ3 in the hippocampus of WT and *Prmt1*+/- mice.

**Figure supplement 2.** Identification of in vivo arginine methylation sites of KCNQ2 MS/MS spectra of the methylated peptides.

**Figure supplement 3.** The sequence alignment of 5 difference human KCNQ isoforms.

**Figure supplement 4.** In vitro methylation assays with GST-Prmt1 or myc-Prmt5 together with GST-KCNQ2 (320-449 aa).

**Figure supplement 5.** Expression of Prmt8 in Prmt1+/+ and Prmt1+/- brain lysates.

residues such as Arg333 (R333), Arg345 (R345), Arg353 (R353), and Arg435 (R435), were identified as methylated in vivo (*Table 1* and *Figure 5—figure supplement 2*). To determine whether Prmt1 methylates KCNQ2, we performed in vitro methylation assays using various bacterially purified GST-Prmts (Prmt1-6) or myc-tagged Prmt5 purified from HEK293T cells and GST-KCNQ2 (amino acids 320–449) that contain all of four arginine residues identified by mass spectrometry. GST-KCNQ2 was methylated by Prmt1, but not by other Prmts (*Figure 5e* and *Figure 5—figure supplement 4*). We next examined whether the newly identified methylation sites were direct targets for methylation by Prmt1. KCNQ2 mutants in each of which either R333, R345, R353, or R435 were substituted with lysine (further referred as RK mutants) were generated and analyzed by in vitro methylation assay with Prmt1 (*Figure 5f*). While the single arginine-to-lysine substitutions did not result in observable decrease of methylation (data not shown), the reduction in KCNQ2 methylation was readily detected with the double RK mutants, R333K/R353K, R345K/R353K, or R345K/R435K. This was further declined in the triple mutant of R333K/R345K/R353K, compared to double mutants. Furthermore, substitution of the four Arg's in KCNQ2 (4RK, R333K/R345K/R353K/R435K) completely abolished the methylation by Prmt1, suggesting that these four arginine residues are critical targets for Prmt1-dependent methylation.

To test whether Prmt1 methylates KCNQ2 in vivo, HEK293T cells expressing HA-KCNQ2 were transfected with control or *Prmt1* shRNA, followed by immunoprecipitation with anti-KCNQ2 antibodies and immunoblotting with antibodies to asymmetric dimethylarginine (ADMA) and KCNQ2. *Prmt1* knockdown specifically decreased the ADMA-positive KCNQ2 levels, suggesting that KCNQ2 methylation is sensitive to Prmt1 levels (*Figure 5g*). Conversely, Prmt1 overexpression enhanced KCNQ2 methylation (*Figure 5h*). We next examined the arginine methylation status of KCNQ2 channels in WT or *Prmt1*+/- brains (*Figure 5i*). The immunoprecipitation analysis showed that ADMA-positive KCNQ2 proteins were readily detected in the WT brain, which were significantly reduced in that of *Prmt1*+/-. These results strongly advocate that Prmt1 interacts with and methylates KCNQ2. Prmt1 and Prmt8 often share substrates in vitro, and PRMT8 is neuron-specific (*Kousaka et al., 2009*). We analyzed whether Prmt8 contributes to hypo-methylation of KCNQ2 proteins in *Prmt1*+/- brains. The protein expression of Prmt8 in the *Prmt1*+/- brain was unaltered from WT brain (*Figure 5—figure supplement 5*). Thus, Prmt8 might not be involved in the decreased KCNQ2 methylation in *Prmt1*+/- brain. Taken together, these data indicate that reduced Prmt1 levels cause hypo-methylation of KCNQ2 in the *Prmt1*+/- brain.

## Methylation regulates the KCNQ2 channel activity

To examine the functional role of Prmt1-mediated methylation of KCNQ2, KCNQ2 channel activities were assessed in *Prmt1*-knockdown HEK293T cells (*Figure 6a–b*, *Figure 6—source data 1*). Using the conventional whole-cell patch clamp technique, the channel function was measured by applying 'step' pulses from –70 to +40 mV in 10-mV increments at a holding potential of –60 mV for 1 s, followed by a tail pulse to –60 mV. *Figure 6a* demonstrates representative whole-cell current traces recorded from KCNQ2-transfected HEK293T cells. Consistent with previous studies (*Lee et al., 2010*), KCNQ2-transfected cells displayed slowly activating outward currents and tail currents. In non-transfected cells, we observed small endogenous outward currents at positive voltage step pulses, but no tail currents (data not shown). The shRNA-mediated depletion of *Prmt1* decreased KCNQ current density from 57.6 ± 7.6 pA/pF of control transfected cells to 26.3 ± 7.6 pA/pF. Rectification of the *I*-V curves was not changed by *Prmt1* knockdown. To verify the role of Prmt1-

**Table 1.** Mono- and dimethylated peptides identified from KCNQ2.

| Peptide* | Modification | Charge state |
|---|---|---|
| [332]RRNPAAGLIQSAWR[345] | Mono-methylated R333 | +3 |
| [334]NPAAGLIQSAWR[345] | Mono-methylated R345 | +2 |
| [346]FYATNLSR[353] | Di-methylated R353 | +2 |
| [435]RSPSADQSLEDSPSK[449] | Di-methylated R435 | +3 |

*Methylated amino acid residues are highlighted in bold.

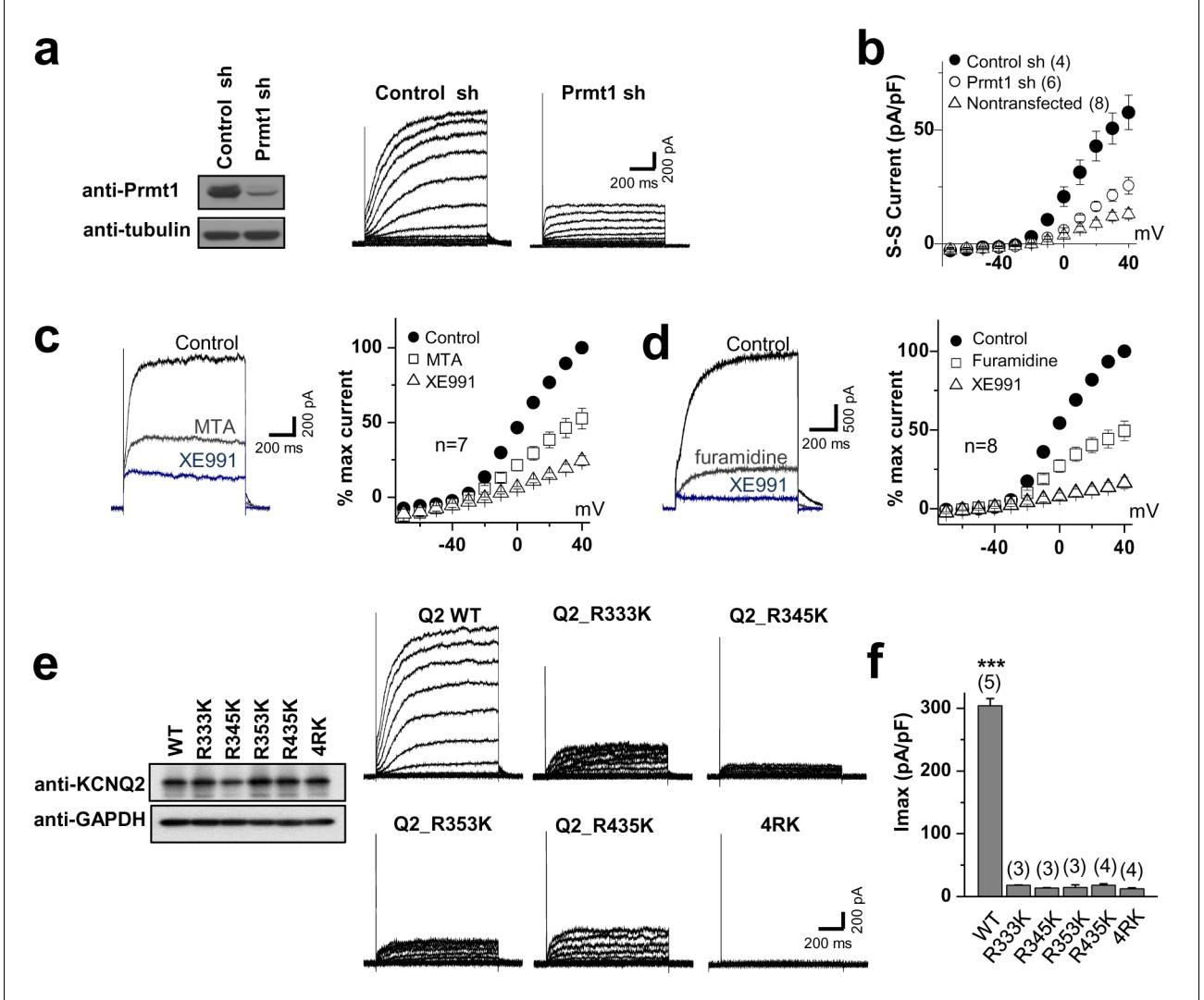

**Figure 6.** Methylation of KCNQ2 regulates its channel activity. (a) Representative current recordings from HEK293T cells expressing KCNQ2 with control shRNA vector (left) or a *Prmt1* shRNA (*Prmt1* sh) (right). Currents were elicited by voltage steps from -70 mV to +40 mV with a subsequent step to -60 mV. (inset) Control immunoblotting for Prmt1 expression. (b) I-V relationships in control cells (•) or *Prmt1* knockdown cells (○) or nontransfected cells (△) are shown. (c–d) Treatment of MTA (c) or furamidine (d) induces a reduction of KCNQ2 currents (left). Both of them suppress KCNQ2 currents without altering shape of I-V curves (right). XE991, KCNQ channel blocker. (e) Representative current recordings from cells expressing WT and methylation defective mutants. (inset) The control western blot shows the expression of various KCNQ vectors. (f) The current density of KCNQ2 at +40 mV for each mutant shown in **e**. ANOVA Tukey test, ***p<0.001.

The following source data is available for figure 6:

**Source data 1.** Source data for *Figure 6*.

mediated methylation in KCNQ2 channel function, the effect of Prmt1 inhibitors on KCNQ2 current was examined. In agreement with *Prmt1* knockdown data, the treatment of MTA or furamidine dramatically reduced the KCNQ2 currents (*Figure 6c–d*, *Figure 6—source data 1*). The KCNQ2 currents were confirmed by XE991 treatment. Assuming that the XE991-sensitive portion entirely represents KCNQ2 currents, 100 μM MTA or 20 μM furamidine inhibited KCNQ2 currents by 79.9 ± 13.6% and 72.1 ± 2.1%, respectively.

We further examined the correlation between the Prmt1-induced methylation and KCNQ2 channel function by recording whole-cell currents in HEK293T cells expressing WT or KCNQ2 mutants. Compared to WT, the KCNQ2 mutants with reduced methylation exhibited remarkably decreased

KCNQ2 currents (*Figure 6e*). The current density at +40 mV was 304.4 ± 11.0 pA/pF (*n* = 5) for WT KCNQ2 channels. We observed significantly decreased KCNQ2 currents in cells expressing RK mutants (R333K, R345K, R353K, R435K, or 4RK): in these cells the current density (pA/pF) of steady-state currents at +40 mV were 18.2 ± 0.3 (*n* = 3, p<0.001), 13.7 ± 0.5 (*n* = 3, p<0.001), 14.8 ± 3.9 (*n* = 3, p<0.001), 18.0 ± 2.2 (*n* = 4, p<0.001), and 12.5 ± 1.6 (*n* = 4, p<0.001), respectively (*Figure 6f*, *Figure 6—source data 1*). These data suggest that the arginine methylation of KCNQ2 is required for its channel activity. Importantly, KCNQ2 mutants showed no alteration in sensitivity to retigabine, KCNQ channel activator (Figure 9d–g, summary data in Figure 9i, *Figure 9—source data 1*), indicating that methylation regulates channel function through a distinct molecular mechanism not overall potential of channel activation.

## Arginine methylation of KCNQ2 is required for maintaining PIP$_2$ affinity

To assess whether methylation of KCNQ2 regulates its cell surface localization, we labeled total plasma membrane proteins of intact cells with membrane-impermeable biotinylation reagents, and isolated membrane proteins with streptavidin beads. The ratio of channels on the membrane to those in the total lysates was not significantly different between control cells and *Prmt1* knockdown cells (*Figure 7—figure supplement 1*, left panel). KCNQ2-4RK also showed a similar amount of biotinylation when expressed in HEK293T cells (*Figure 7—figure supplement 1*, right panel). The absence of biotinylation on the cytosolic marker protein heat shock protein-90 confirmed specificity of the surface labeling. As the surface-resident channels remained unchanged, the KCNQ conductance should be determined by the open probability.

KCNQ/M channels require a certain level of PIP$_2$ in the cell membrane to maintain their activity (*Delmas and Brown, 2005*; *Suh and Hille, 2002*; *Winks et al., 2005*; *Zhang et al., 2003*). PIP$_2$ acts to stabilize the open state of KCNQ2 channels resulting in increased open probability at all voltage levels. Thus, the depletion of PIP$_2$ (*Suh and Hille, 2002*; *Winks et al., 2005*; *Zhang et al., 2003*) or the reduction of PIP$_2$ affinity of channel (*Kosenko et al., 2012*; *Lee et al., 2010*; *Hernandez et al., 2008*) can suppress KCNQ currents. Interestingly, the methylated arginine residues, R333, R345, R353, and R435, reside in the PIP$_2$-binding domain of KCNQ2 (*Suh and Hille, 2008*; *Hernandez et al., 2008*). This led us to postulate that methylation might affect the PIP$_2$ binding affinity of KCNQ2 thereby regulating channel activities.

To appraise the affinity of PIP$_2$ for KCNQ2, we utilized neomycin, which is widely used to sequester PIP$_2$ (*Liscovitch et al., 1994*; *Haider et al., 2007*; *Suh and Hille, 2007*). Since neomycin is a polycation that binds specifically to PIP$_2$, it has been used to determine the PIP$_2$ content in biological membranes (*Arbuzova et al., 2000*). The neomycin sensitivity of ion channels such as Kir and KCNQ is well regarded as a measure of its PIP$_2$ affinity (*Haider et al., 2007*; *Suh and Hille, 2007*; *Kosenko et al., 2012*; *Schulze et al., 2003*). Accordingly, ion channels with high PIP$_2$ affinities are expected to be less sensitive to neomycin than those with low PIP$_2$ affinity. Inclusion of 10 μM neomycin in the patch pipette solution led to a moderate inhibition of the WT KCNQ2 current in HEK293T cells (*Figure 7a*, *Figure 7—source data 1*). In contrast, mutant channels with reduced methylation showed greatly increased inhibition by 10 mM neomycin. So, the remaining current at 10 min time points after rupturing the membrane is significantly reduced from 81.2 ± 1.9% (WT, n = 5) to 48.7 ± 4.2% (R333K, n = 5, p<0.01; *Figure 7a*). We measured dose-responses to neomycin. In comparison with WT, methylation-deficient mutant KCNQ2 (R333K, R345K, R353K, R435K) exhibited a higher sensitivity to neomycin (*Figure 7b*, *Figure 7—source data 1*), suggesting lower PIP$_2$ affinity. To directly examine whether these channels have different PIP$_2$ sensitivity, we included 20 μM or 200 μM diC8-PIP$_2$ in the patch pipette solution and measured current augmentation after rupturing the plasma membrane (*Figure 7c and d*, *Figure 7—source data 1*). WT KCNQ2 did not show increase in current by diC8-PIP$_2$ even at 200 μM, suggesting that endogenous PIP$_2$ might be at a saturating concentration. In contrast, methylation-deficient mutant KCNQ2 (R333K, R345K, R353K, R435K, 4RK) showed a dose-dependent increase in current but they did not reach to WT level at corresponding concentrations of PIP$_2$. Only R333K reached to the WT level when introduced to 200 μM PIP$_2$. On the other hand, 4RK appeared to be most severely affected in PIP$_2$ binding. Altered PIP$_2$ affinity in RK mutants was also assessed by a voltage-sensitive phosphatase from Danio rerio (Dr-VSP), which hydrolyzes PIP$_2$ at highly depolarized voltages (e.g., +100 mV) and transiently reduces the PIP$_2$ level (*Falkenburger et al., 2010*; *Rjasanow et al., 2015*). Dr-VSP was coexpressed with WT KCNQ2 or RK mutants and its activity was elicited by membrane depolarization. Consistent with a

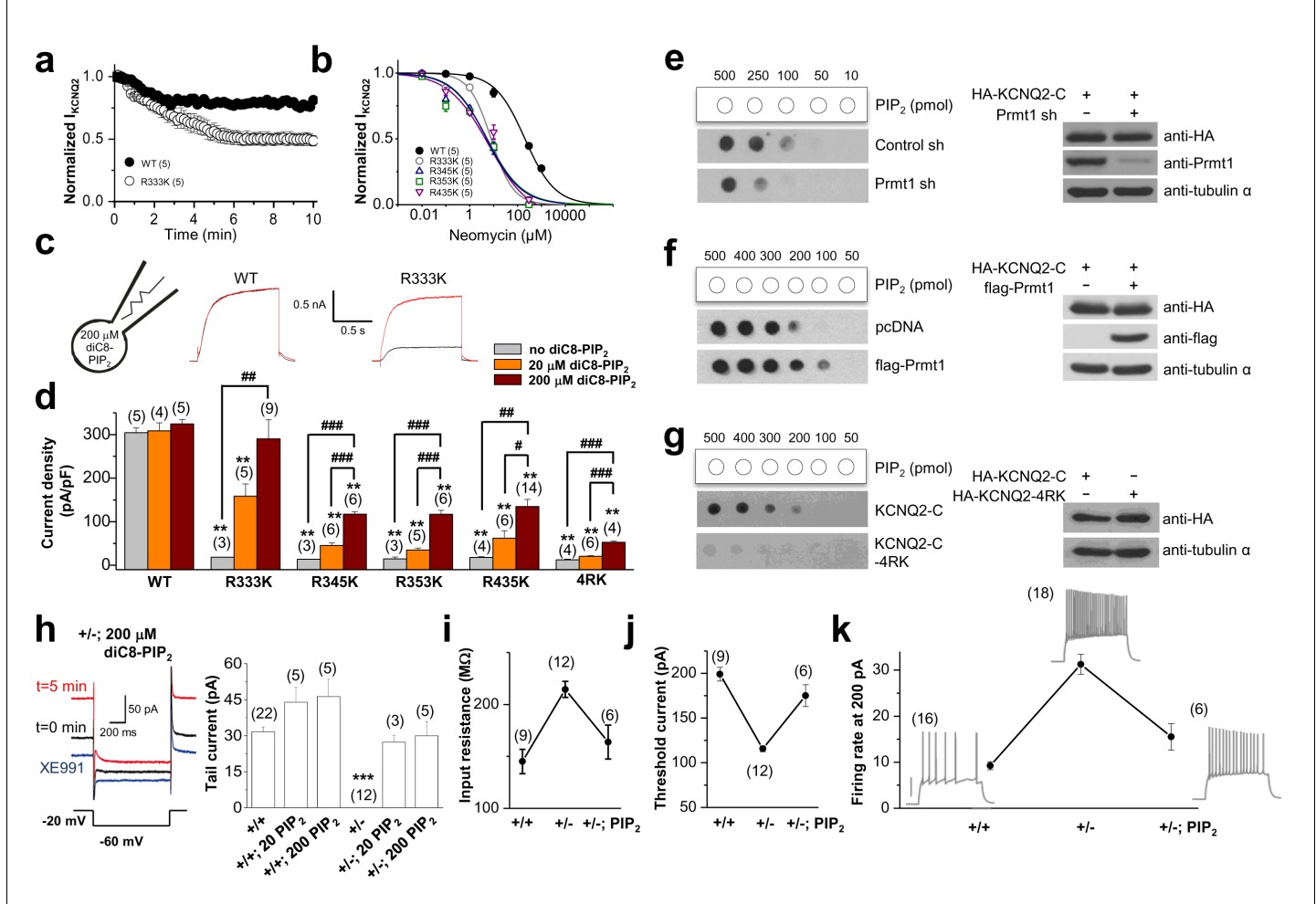

**Figure 7.** Methylation of KCNQ2 determines its PIP$_2$ affinity. (**a**) Pooled data show 10 μM neomycin-induced rundown of WT (•) and R333K (○) KCNQ2 currents. KCNQ2 currents were normalized to KCNQ2 current at t = 0. (**b**) Dose-response curves for neomycin measured at 10 min after rupturing the plasma membrane. Indicated concentration of neomycin was included in the patch pipette. Solid lines are Hill fits to the mean data for WT, R333K, R345K, R353K, and R435K giving IC50 values of 59.1 ± 8.3, 9.2 ± 4.2, 5.2 ± 0.2, 5.4 ± 3 and 6.2 ± 8 μM and slopes of 0.8 ± 0.05, 0.8 ± 0.3, 0.6 ± 0.1, 0.6 ± 0.2, and 0.5 ± 0.2, respectively. (**c**) Representative traces showing a significant increase in R333K KCNQ2 mutant by 200 μM diC8-PIP$_2$ addition to patch pipette (red line) compared to controls (black line). KCNQ2 (WT) did not show apparent augmentations. Currents were elicited by voltage steps from -60 mV to +40 mV. (**d**) from data such as shown in c current densities for 0, 20, and 200 μM diC8-PIP$_2$ in the recording pipette were determined and plotted as bars with S.E.M. Currents were measured at >20 min after rupturing the plasma membrane. The numbers in parentheses indicate the number of cells. ANOVA Tukey test. **$p<0.01$ versus corresponding concentration to WT. #$p<0.05$; ##$p<0.01$; ###$p<0.001$. (**e–g**) Effects of Prmt1 (e-f) or 4RK mutation (g) on the binding of KCNQ2 and PIP$_2$. Different amounts (10–500 pmol) of PIP$_2$ were spotted onto nitrocellulose membranes and analyzed by a protein-lipid overlay procedure using cell lysate prepared from HEK293T cells transfected with indicated expression vectors or the control pcDNA vector (left panels). The control western blots are shown in right panels. (**h**) Left, representative traces show augmentation of M-current by 200 μM diC8-PIP$_2$ in the patch pipette. Right, summary of M-current density from WT neurons and *Prmt1+/-* neurons with 0, 20, and 200 μM diC8-PIP$_2$ in the recording pipette. Mean ± S.E.M. ANOVA Tukey test, ***$p<0.001$. (**i–j**) the mean value of input resistance (i), and threshold current for AP generation (100 ms duration; j) from WT (+/+) neurons, mutants (+/-), and mutants loaded with 20 μM diC8-PIP$_2$ (+/-; PIP$_2$). (**k**) the mean number of APs in response to 1-s depolarizing current injection (200 pA) from a WT (+/+) neuron, a mutant (+/-), and a mutant loaded with 20 μM diC8-PIP$_2$ (+/-; PIP$_2$).

The following source data and figure supplements are available for figure 7:

**Source data 1.** Source data for *Figure 7*.
**Figure supplement 1.** Surface expression of KCNQ channels was not affected by *Prmt1* knockdown or 4RK mutation.
**Figure supplement 2.** Quantitative determination of the sensitivity of KCNQ2 channels to activation of Dr-VSP in HEK293T cells.

*Figure 7 continued*

**Figure supplement 2—source data 1.** Source data for *Figure 7—figure supplement 2*.

**Figure supplement 3.** Comparison of XE991 sensitivity in GCs.

**Figure supplement 3—source data 1.** Source data for *Figure 7—figure supplement 3*.

**Figure supplement 4.** Application of exogenous PIP2 had little effect on the excitability in WT neurons.

**Figure supplement 4—source data 1.** Source data for *Figure 7—figure supplement 4*.

**Figure supplement 5.** The effects of SK channel block and BK channel block on neuronal excitability of control or furamidine-pretreated GCs.

**Figure supplement 5—source data 1.** Source data for *Figure 7—figure supplement 5*.

previous report (*Falkenburger et al., 2010*), activation of Dr-VSP reduced WT KCNQ2 currents and currents were recovered quickly after $PIP_2$ resynthesis on repolarization (*Figure 7—figure supplement 2*, *Figure 7—figure supplement 2—source data 1*). When subjecting RK mutants to the same VSP activation protocol, the overall behavior was similar to the WT KCNQ2 channel; however, the recovery of current was slowed 2–4 folds (*Figure 7—figure supplement 2*, *Figure 7—figure supplement 2—source data 1*). The mean time to the 70% maximum current ($T_{70}$) for R333K, R345K, R353K and R435Kwas 22.8 ± 3.7, 24.1 ± 1.9, 32.1 ± 2.6, and 17.9 ± 1.5 s, respectively (p<0.05 vs WT:9.5 ± 3.1 s). The slowed recovery after VSP reflects the reduced $PIP_2$ affinity (*Rjasanow et al., 2015*), further supporting for the reduced $PIP_2$ affinity of RK mutants. Taken together, these data suggest that methylation of KCNQ2 channel regulates its interaction with $PIP_2$.

To further confirm that arginine methylation modulates the $PIP_2$ binding affinity of KCNQ2, we used a protein-lipid overlay (PLO) assay with an HA-KCNQ2-C (*Figure 7e–g*). $PIP_2$ strips were incubated with lysates from control or *Prmt1* knockdown cells expressing HA-KCNQ2-C. We found that the $PIP_2$ affinity of KCNQ2 decreased in the *Prmt1* knockdown cells compared to that in the control cells (*Figure 7e*). In contrast, overexpression of *Prmt1* with HA-KCNQ2 enhanced the affinity of KCNQ2 for $PIP_2$ (*Figure 7f*). Compared to that of WT KCNQ2, methylation-deficient mutant channels also had very little affinity for $PIP_2$ (*Figure 7g*), consistent with electrophysiological measurements.

We then tested whether $PIP_2$ loading could also rescue M-currents in neurons of *Prmt1* +/- mice. As shown in *Figure 7h,M*-current density in *Prmt1*+/- neurons was restored to the WT level by $PIP_2$ loading. Introduction of 20 and 200 µM diC8-$PIP_2$ to *Prmt1*+/- GCs increased M-currents to 27.3 ± 2.9 (n = 3) and 29.9 ± 5.9 pA (n = 5), respectively, which are insignificantly different from that of WT (31.65 ± 2.03 pA, n = 22, p>0.05; *Figure 7h*, *Figure 7—source data 1*), while exogenous $PIP_2$ had little effects on M-currents in WT GCs (p>0.05, *Figure 7h*, *Figure 7—source data 1*). As expected from opening of the M-channels by $PIP_2$, the input resistance decreased significantly from 214.8 ± 7.7 MΩ to 163.8 ± 16.4 MΩ (*Figure 7i*, *Figure 7—source data 1*), and the AP threshold current increased from 115.6 ± 3.3 pA to 175 ± 12.1 pA (*Figure 7j*, *Figure 7—source data 1*). These results indicate that declined methylation caused by *Prmt1* depletion impairs KCNQ channel activity via reduction in channel-$PIP_2$ interaction.

Since $PIP_2$ restored the diminished KCNQ currents, we asked whether $PIP_2$ could also reduce the observed neuronal hyperexcitability in *Prmt1*+/- neurons. We found that $PIP_2$ loading decreased the firing rate toward normal values in *Prmt1*+/- GCs. For example, the AP frequency of *Prmt1*+/- GCs in response to a 200 pA depolarizing current was 31.2 ± 2.2 Hz, and decreased to 15.5 ± 2.9 Hz following $PIP_2$ loading (*Figure 7k*, *Figure 7—source data 1*). In addition, *Prmt1*+/- neurons also regained the sensitivity to XE991 (*Figure 7—figure supplement 3*, *Figure 7—figure supplement 3—source data 1*). In contrast, $PIP_2$ application had little effects on the excitabilities of WT neurons (*Figure 7—figure supplement 4*, *Figure 7—figure supplement 4—source data 1*). These results suggest that the impaired KCNQ channel function in the hippocampus of *Prmt1*+/- mice can be

restored by PIP$_2$ addition. Although the recovery of excitability with PIP$_2$ application was significant it was incomplete, suggesting the possible involvement of other Prmt1 target(s) in neuronal hyperexcitability. As immediate candidates, we tested other K$^+$ channels such as SK and BK channels in Prmt1 effects which are known to regulate membrane excitability of dentate gyrus GCs (*Brenner et al., 2005*). The data obtained by using apamin and paxilline, specific SK and BK channel blockers, respectively, ruled out that the contribution of these channels in hyperexcitability of *Prmt1*-deficient GCs (*Figure 7—figure supplement 5*, *Figure 7—figure supplement 5—source data 1*). Further study will be required to identify additional targets of Prmt1. Taken together, genetic deletion or pharmacological inhibition of Prmt1 reduces the affinity between the KCNQ channel and PIP$_2$, leading to reduction of KCNQ currents accompanied by neuronal hyperexcitability.

## Protein arginine methylation is critical for cytoprotective neuronal silencing

Prmt1 seems to play a key role in physiological functions of KCNQ channels by controlling its interaction with PIP$_2$ at the basal state. KCNQ channels are also known to contribute to cellular protections in pathologic condition. In particular, the enhancement of KCNQ currents in response to oxidative stress led to a dramatic reduction of the AP firing frequency that may prevent neuronal death (*Patel, 2004*). We then examined whether Prmt1-mediated methylation is functionally associated with the mechanisms of cytoprotective enhancement of KCNQ currents under oxidative stress.

To investigate the role of Prmt1 in reactive oxygen species (ROS)-induced augmentation of KCNQ channel and consequential neuronal silencing, neuronal excitabilities in WT and *Prmt1*+/- GCs are evaluated in response to H$_2$O$_2$. Consistent with previous studies (*Gamper et al., 2006*), H$_2$O$_2$ treatment strongly reduced neuronal excitabilities in WT GCs (*Figure 8a–b*). The AP firing rate during injection of a 200 pA depolarizing current was 9.3 ± 2.1 Hz, and decreased to 1.4 ± 0.9 Hz (*Figure 8c*, *Figure 8—source data 1*) after H$_2$O$_2$ treatment. Consistent with the KCNQ current augmentation, the input resistance was decreased from 163.8 ± 24.9 to 116.3 ± 21.8 MΩ (*Figure 8d*, *Figure 8—source data 1*). In contrast, H$_2$O$_2$ application did not lower firing rates and input resistance in GCs of *Prmt1*+/- mice, although their firing rates and input resistance were elevated compared to WT (*Figure 8e–h*, *Figure 8—source data 1*). Consistently, the pharmacological inhibition of arginine methylation with MTA (*Figure 8i–l*, *Figure 8—source data 1*) or furamidine (*Figure 8m–p*, *Figure 8—source data 1*) also blocked neuronal silencing and input resistance reduction in WT GCs in response to the oxidative stress inflicted by H$_2$O$_2$. These data indicate that Prmt1-mediated protein arginine methylation is necessary for ROS-induced neuronal silencing via KCNQ channel activation.

Consistent with a previous study (*Gamper et al., 2006*), H$_2$O$_2$ induced a sharp augmentation of WT KCNQ2 channels heterologously expressed in HEK293T cells (*Figure 9a*). The mean current augmentation induced by 500 µM H$_2$O$_2$ was 2.6 ± 0.2-fold (*Figure 9h*). At the concentration of 500 µM, the H$_2$O$_2$ effect usually reached a plateau within 7~8 min of application. The H$_2$O$_2$ effect was completely reversed by the addition of a reducing agent, dithiothreitol (DTT, 2 mM), suggesting that this effect is due to a reversible oxidative modification. Notably, DTT-reduced KCNQ2 currents in H$_2$O$_2$-treated cells were back to the resting level within several minutes, but did not inhibit them further (*Figure 9a*). Also, DTT had no effect on KCNQ2 currents in non- H$_2$O$_2$-treated cells (data not shown). When protein arginine methylation was blocked with the treatment of MTA (*Figure 9b*) or furamidine (*Figure 9c*), KCNQ2 channels appeared to be insensitive to H$_2$O$_2$ with the currents augmented by H$_2$O$_2$ (500 µM) by 0.9 ± 0.02-fold and 0.8 ± 0.03-fold, respectively (*Figure 9h*, *Figure 9—source data 1*). We then examined whether the sensitivity to H$_2$O$_2$ is abrogated by the KCNQ2 mutants with reduced methylation. As shown in *Figure 9d–g*, the mutant channels were insensitive to H$_2$O$_2$, compared to WT KCNQ2. Currents of the R333K, R345K, R353K and R435K mutant channels were augmented by 1.6 ± 0.3-, 0.9 ± 0.1-, 0.7 ± 0.1- and 1.1 ± 0.1-fold, respectively (*Figure 9h*, *Figure 9—source data 1*).

Recently, Gamper et al. (*Gamper et al., 2006*) showed that a triple cysteine pocket in the S2-S3 linker is critical for the H$_2$O$_2$-induced enhancement of KCNQ channel. Interestingly, MTA and furamidine reduced activities of the triple-Cys mutant of KCNQ2 by a similar extent as it did those of the WT KCNQ2 channel, while the mutant was insensitive to H$_2$O$_2$ as expected (*Figure 9—figure supplement 1*, Figure 9—figure supplement 1—source data 1). This result indicates that the H$_2$O$_2$ activation pathway is unnecessary for the arginine methylation-mediated regulation of Prmt1. Taken

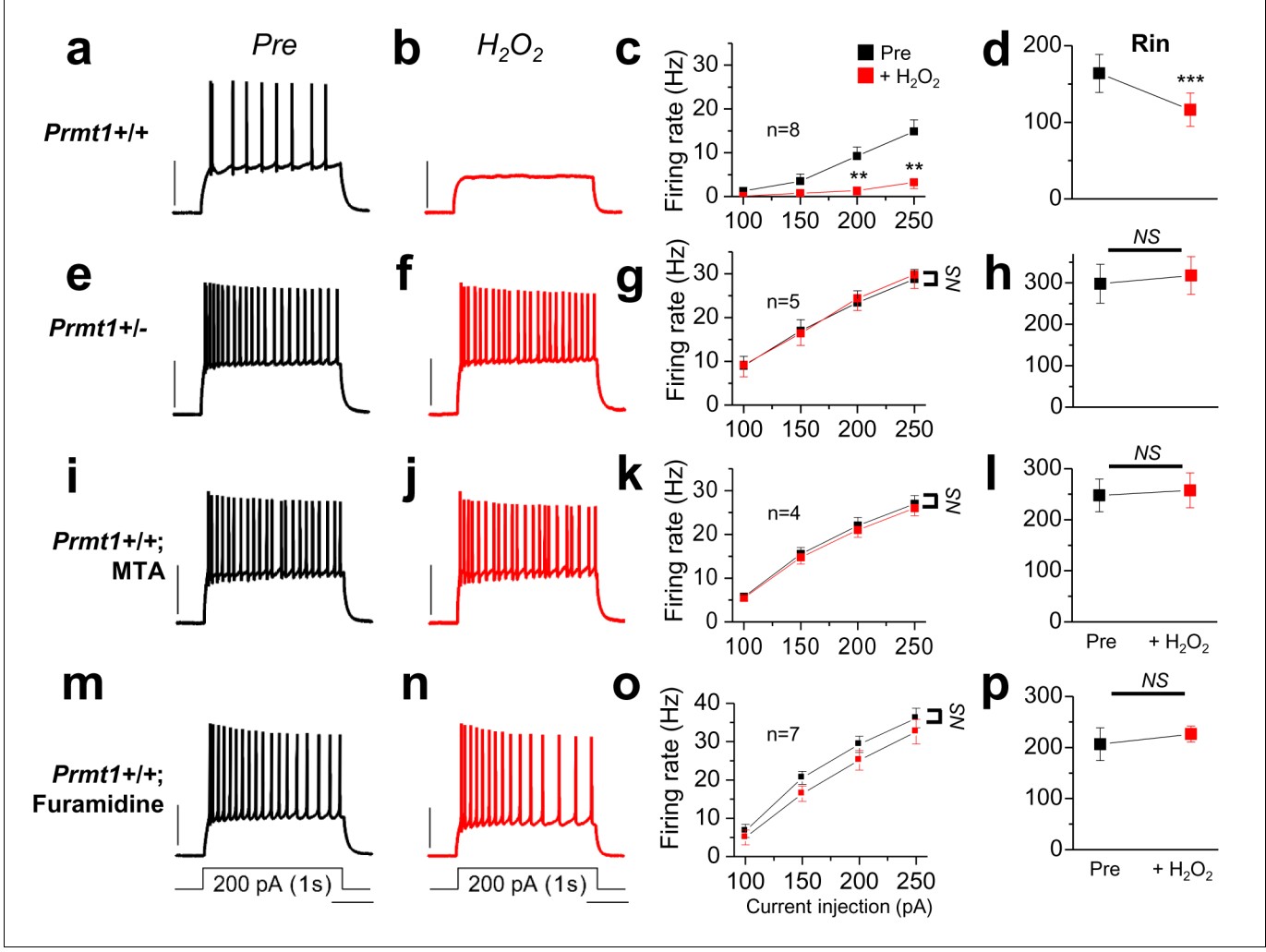

**Figure 8.** Prmt1 is involved in Oxidative stress-induced silencing of neurons. Spike trains were evoked by injecting 1-s depolarizing current pulses of different intensities (100–250 pA) into the cell before (black) and after (red) application of $H_2O_2$. Left panels show representative recordings in normal extracellular medium (**a,e,i,m**) and middle panels after application of 500 µM $H_2O_2$ for 10 min (**b,f,j,n**). $H_2O_2$ had little effect on firing rates in *Prmt1+/-* GCs (**f**; *n* = 5, *four mice*), MTA (**j**; *n* = 4, *four mice*)- or furamidine (**n**; *n* = 7, *four mice*)-treated WT GCs than in untreated WT GCs (**b**; *n* = 8, *four mice*). Summary diagrams (right column) compare firing rates or input resistance before (black) and after (red) application of $H_2O_2$ in WT (**c,d**), *Prmt1+/-* (**g,h**), MTA (**k,l**)- or furamidine (**o,p**)-treated WT dentate GCs. **p<0.01; ***p<0.001 by Student's *t*-test.

The following source data is available for figure 8:

**Source data 1.** Source data for *Figure 8*.

together, Prmt1 is prerequisite for cytoprotective enhancement of KCNQ currents in the milieu of oxidative stress.

## Discussion

In this study, we describe a novel regulatory mechanism whereby methylation of KCNQ2 affects KCNQ/M channel activities and overall excitabilities of the brain. The heterozygous reduction of the *Prmt1* gene dose in mice caused spontaneous seizures. The elevated neuronal excitability in *Prmt1* +/- mice appears to be a direct result of reduced KCNQ/M channel activities in that KCNQ2 from the *Prmt1+/-* brain was less asymmetrically methylated and showed diminished affinity to $PIP_2$. Furthermore, the activation of KCNQ channel under oxidative stress as well as its basal activity at rest

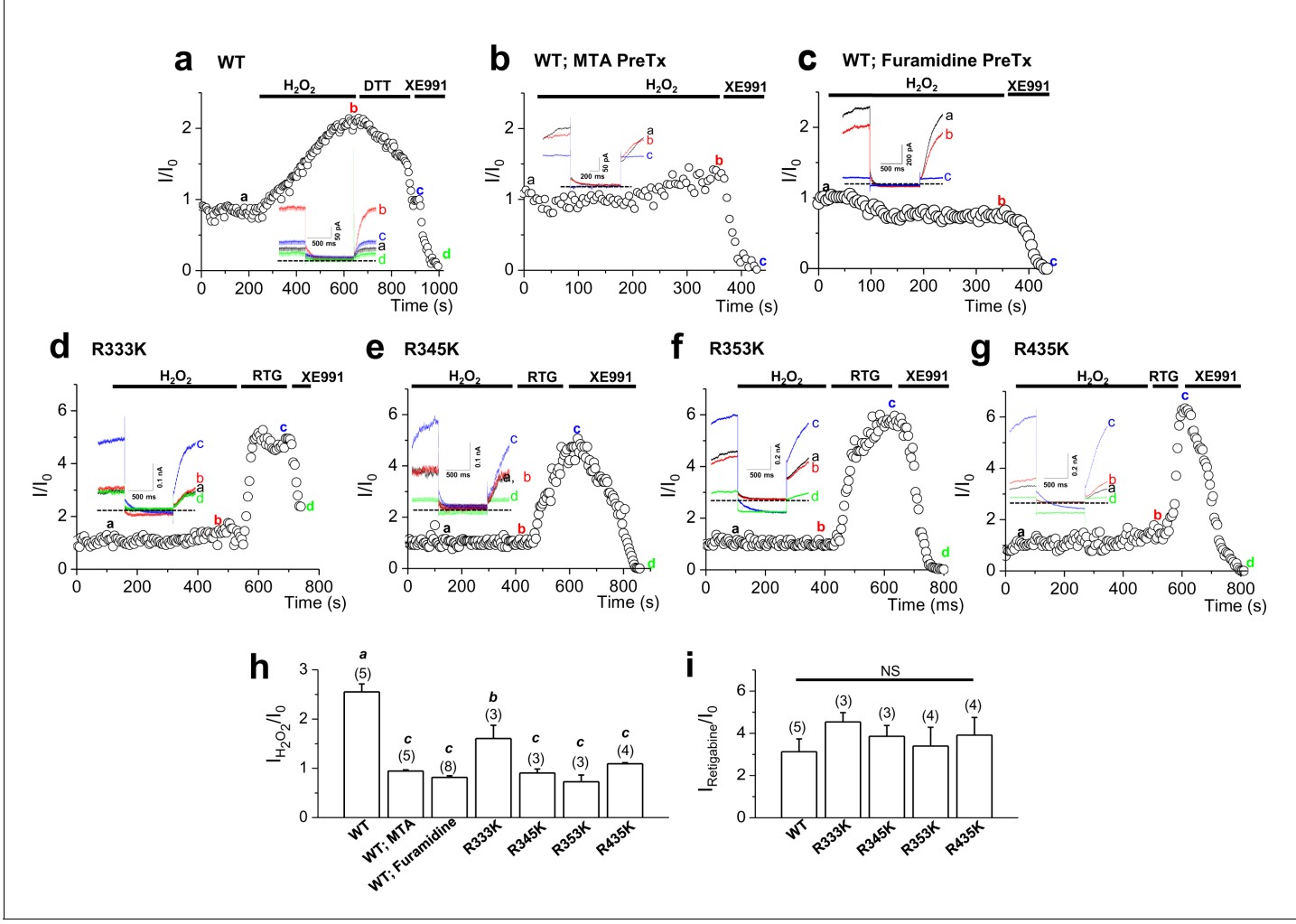

**Figure 9.** Activation of KCNQ2 channels by $H_2O_2$ is sensitive to methylation. (a–g) Time course for the effect of 500 μM $H_2O_2$ on KCNQ2 currents. WT KCNQ2 (a-c), R333K (d), R345K (e), R353K (f) or R435K (g) were expressed in HEK293T cells. For experiment (b) & (c), cells were pretreated with MTA (100 μM) or furamidine (20 μM), respectively. Whole-cell currents were monitored by 1-s hyperpolarizing steps to -60 mV from a holding potential of -20 mV at 5-s intervals. $H_2O_2$ (500 μM), DTT (2 mM), the M-channel blocker, XE991 (50 μM), and M-channel activator, retigabine (RTG; 10 μM), were applied during the periods indicated by the bars. Plotted are normalized current amplitudes versus time during the experiment. Representative current traces from each experiment (not normalized) are depicted in the insets. (h) Summarized data for KCNQ2 activation induced by $H_2O_2$. Letters indicate statistically distinct groups (ANOVA Tukey, p<0.01). (i) Summarized data for KCNQ2 activation induced by retigabine.

The following source data and figure supplements are available for figure 9:

**Source data 1.** Source data for *Figure 9*.

**Figure supplement 1.** Time course for the effect of 500 μM $H_2O_2$ on Triple-Cys mutant of KCNQ2 currents (left).

**Figure supplement 1—source data 1.** Source data for *Figure 9—figure supplement 1*.

depends on protein arginine methylation. It is well known that ROS generated under oxidative stress induces augmentation of KCNQ currents in neurons (*Gamper et al., 2006*). The resulting reduction in neuronal discharge lowers electrical activity of the cell and energy consumption, thereby providing an important mechanism of protecting neurons from the oxidative stress-induced death (*Gamper et al., 2006*; *Won et al., 2002*). Considering that oxidative stress is also involved in induction of seizures (*Patel, 2004*), the impairment of neuroprotective KCNQ activation in response to oxidative stress, may contribute to spontaneous seizures together with the reduced basal activity of

KCNQ channels in *Prmt1*+/- mice. Protein arginine methylation is a unique post-translational modification that increases the KCNQ/M channel function unlike other known regulatory signaling pathways, and thus provides a platform for the design of novel therapeutic strategies for epilepsy and other neuronal hyperexcitability disorders.

Our results find that 4 arginine residues (R333, R345, R353, and R435) are the major methylation sites in KCNQ2. Interestingly, all of the methylation sites reside within the C-terminal PIP$_2$ binding domain. This is known as a 'hot spot', as it serves as binding sites for multiple signaling molecules as well as PIP$_2$ (*Delmas and Brown, 2005*; *Hernandez et al., 2008*; *Zhang et al., 2003*). The observation that decreased methylation lowers PIP$_2$ affinity proposes that Prmt1 methylation on the arginine residues would increase channel-PIP$_2$ interaction. Indeed, previous computational analyses revealed that methylation renders the guanidinium of arginines to be more electron-rich and thus to exhibit higher pKa values (*Shearer, 2008*), which would stabilize the positive charge of the arginine residue in proteins in the physiological condition and thereby likely facilitate their electrostatic interaction with negatively charged molecules. Thus, it is conceivable that arginine methylation is required for the KCNQ channel activity because it promotes electrostatic binding of KCNQ proteins to PIP$_2$.

Various KCNQ channel mutations associated with epilepsy have been shown to cause a reduction in M-channel activity, which all can lead to membrane depolarization and increased neuronal firing via diverse mechanisms. Mutations in the C-terminal region of KCNQ channels reduce channel activities by interfering with channel targeting to the surface membrane (*Schwake et al., 2000*), protein stability (*Soldovieri et al., 2006*), interaction with calmodulin (*Jentsch, 2000*), or modulation by cAMP (*Schroeder et al., 1998*). Mutations affecting the pore region of KCNQ2 may reduce currents by affecting ion channel conductance, while mutations at the S4 domain affect channel gating and increase the threshold for channel activation (*Maljevic et al., 2010*). It is interesting in this context that R353G, a mutant of one of the major methylation sites identified in this study, is linked to familial epilepsy, likely due to reduced calmodulin binding and a consequent defect in membrane trafficking (*Jentsch, 2000*; *Etxeberria et al., 2008*). However, a recent study proposed an additional pathophysiological mechanism involving a diminished PIP$_2$ affinity of the channel for this epileptic KCNQ2 mutation (*Kosenko et al., 2012*). This is in agreement with our current study indicating that disruption of KCNQ2 channel-PIP$_2$ interaction caused by decreased methylation is a key mechanism linked to neuronal hyperactivity.

*Like KCNQ2*, KCNQ3 channel contains conserved methylation sites within the PIP$_2$ biding domain (*Figure 5—figure supplement 3*) suggesting that arginine methylation might be also involved in regulation of channel-PIP$_2$ interaction in KCNQ3 channels as well. Considering that the KCNQ2 and KCNQ3 complex produces M-currents, hypo-methylation of either KCNQs might significantly diminish M-currents and neuronal hyperexcitability. The positively charged arginines are also important for electrostatic interaction with PIP$_2$ in other PIP$_2$-sensitive channels and transporters such as Kir2 (*Hansen et al., 2011*; *Huang et al., 1998*; *Lopes et al., 2002*), and GIRK (*Whorton and MacKinnon, 2011*). PIP$_2$ is required for proper function of many plasma membrane ion channels and transporters (*Suh and Hille, 2008*; *Hilgemann et al., 2001*). Thus, methylation can be a general mechanism for regulating ion flux physiology. Consistent with this, our results show that the effects of KCNQ channel block on firing rates are not the same as those of Prmt1 block (*Figure 3*), implying a possible involvement of other Prmt1 target(s) in neuronal hyperexcitability and seizures. PIP$_2$ loading to *Prmt1*+/- GCs recovered the firing properties significantly but incompletely. It appears that SK and BK channels are not involved in this matter. Thus further studies are required to identify other Prmt1 targets in the control of neuronal excitability. Regardless, the results presented here provide the first direct evidence of functional role for protein arginine methylation in the control of neuronal excitability and also offers a platform to further investigate the physiological and pathophysiological roles of KCNQ/M channel activity in CNS.

## Materials and methods

### Animal studies

*Prmt1*+/- mice were obtained from EUCOMM consortium (C57BL/6N agouti background) and backcrossed onto C57BL/6J background for at least 5 generations before being used for the experiment as previously described (*Choi et al., 2012*). All animal experiments were approved by the

Institutional Animal Care and Research Advisory Committee at Sungkyunkwan University School of Medicine Laboratory Animal Research Center (Approval No. IACUC-11-39). Mice were maintained in C57BL/6J background and bred through heterozygous matings. For this study, we have used only male mice to avoid the complication due to the menstural cycles with female mice.

For EEG surgery and recording, long-term video-EEG monitoring was performed as described previously (Jeon et al., 2011). Male *Prmt1+/-* mice and their littermate WT control mice were anesthetized by intraperitoneal injection of 1% ketamine (30 mg/kg) and xylazine hydrochloride (4 mg/kg). Surgery was performed using a stereotaxic apparatus (Kopf Instruments, USA, California). Recordings were obtained using skull screws (stainless, 1.0 mm in diameter), which were positioned in AP −1.8 mm and L −2.1 mm from the bregma with grounding over the cerebellum. Electrical activities were recorded after being amplified (×1200), bandpass-filtered at 0.1–70 Hz, and digitized with a 400-Hz sampling rate using a digital system (Comet XL, Astro-Med, Warwick, RI, USA). Video-EEG signals were continuously recorded 24 hr per day for 7 days, and the waveforms and epileptiform activities were analyzed offline using Matlab, PSG Twin 4.3 (Astro-Med, USA, West Warwick, RI), and pClampfit 10.2 (Axon Instruments, USA, California). To obtain colored power spectra, EEG signals were filtered from 1 to 70 Hz. Colored power spectra were calculated and drawn with Fourier transformation of 2-s window sizes. The duration of seizures, the number of seizures, the frequency of seizure spikes, and seizure score were measured and analyzed. The seizure score was classified on the basis of Racine's scale (Racine, 1972): stage 0, no changes in behavior; stage 1, arrest, immobility and rigid posture; stage 2, jerking, tail rattling, staring with mouth clonus, and head nodding; stage 3, forelimb clonus; stage 4, rearing with forelimb clonus; stage 5, body shaking, wild running and jumping; stage 6, tonic-clonic seizures. Two *Prmt1+/-* mice only showed interictal-like activities without ictal activities and were removed from analysis.

The open-field task was used to assess locomotor activity (Jung et al., 2013). The open-field box was made of white plastic (40 × 40 × 40 cm) and the open field was divided into a central field (center, 20 × 20 cm) and an outer field (periphery). Male *Prmt1+/-* mice and their littermate WT control mice were used. Individual mice were placed in the periphery of the field and the paths of the animals were recorded with a video camera. The total distance traveled for 30 min and the time spent in the central area for 5 min were analyzed using the program EthoVision XT (Noldus, USA, Virginia).

## Brain slice preparation and recording

Brain slices were prepared from male *Prmt +/-* mice and their littermate WT control mice aged 4–6 weeks old. Mice were killed by decapitation after being anesthetized with pentobarbital sodium, and the whole brain was immediately removed from the skull and chilled in artificial cerebrospinal fluid (aCSF) at 4°C. Transverse hippocampal slices (350 μm thick) were prepared using a vibratome (VT1200S, Leica, Germany, Nussloch). Slices were incubated at 35°C for 30 min and thereafter maintained at 32°C until in situ slice patch recordings and fluorescence microscopy. Hippocampal granule cells of the dentate gyrus were visualized using an upright microscope equipped with differential interference contrast optics (BX51WI, Olympus, Japan, Tokyo). Whole-cell current clamp techniques were used for excitability of dentate granule cells (GCs). The pipette solution contained (in mM): 143 K-gluconate, 7 KCl, 15 HEPES, 4 MgATP, 0.3 NaGTP, 4 Na-ascorbate, and 0.1 EGTA/or 10 BAPTA with the pH adjusted to 7.3 with KOH. The bath solution (or aCSF) for the control experiments contained the following (in mM): 125 NaCl, 25 NaHCO$_3$, 2.5 KCl, 1.25 NaH$_2$PO$_4$, 2 CaCl$_2$, 1 MgCl$_2$, 20 glucose, 1.2 pyruvate, and 0.4 Na-ascorbate, pH 7.4 when saturated with carbogen (95% O$_2$ and 5% CO$_2$). All bath solution included in 20 μM bicuculine and 10 μM CNQX to block the inhibitory synaptic signal. The perfusion rate of the bathing solution and the volume of the recording chamber for slices were 2.2 ml/min and 1.2 ml, respectively. Patch pipettes with a tip resistance of 3–4 MΩ were used. The series resistance ($R_s$) after establishing whole-cell configuration was between 10 and 15 MΩ. Electrophysiological recordings were made in somata with EPC-8 amplifier (HEKA Instruments, Germany, Lambrecht/Pfalz). Experiments were performed at 32 ± 1°C. The following parameters were measured: (1) the resting membrane potential, (2) the input resistance ($R_{in}$, membrane potential changes (V) for given hyperpolarizing current (−35 pA, 600 ms) input), (3) AP threshold current (current threshold for single action potential generation, 100 ms duration), (4) AP height; defined as the peak relative to the most negative voltage reached during the afterhyperpolarization immediately after the spike, (5) AP half-width; measured as the width at half-maximal spike amplitude, (6) F-I curve (firing frequencies (F) against the amplitude of injected currents (I); 100–400 pA). We excluded

data for analysis when series resistance exceeded 20 MΩ or when resting membrane potential was more positive than −60 mV. The whole-cell voltage clamp technique was used to measure K$^+$ currents. Whole-cell K$^+$ currents, evoked in response to voltage step to potentials ranging from –70 mV to +30 mV (in 10 mV increments, 1 s duration) from a holding potential of –60 mV, were examined in WT and *Prmt1*+/- dentate GCs.

## Plasmids

Expression vectors encoding human KCNQ2 and Triple-Cys mutant of KCNQ2 channel were generously provided by Dr. Shapiro (University of Texas Health Science Center) (*Falkenburger et al., 2010*). To construct the HA -tagged KCNQ2 and HA-tagged KCNQ2-C (aa320-840) expression vectors, the entire coding region of KCNQ2 or the corresponding sequence of KCNQ2 aa320-840 were amplified by polymerase chain reaction (PCR) and the PCR products were cloned into pcDNA3.1-HA vector (Clontech, USA, California). A series of KCNQ2 arginine-to-lysine mutants were generated by site-directed mutagenesis using a QuickChange kit (Stratagene, La Jolla, USA, California). To generate GST fusion protein of KCNQ2 aa320-449, the corresponding region was amplified and cloned into PGEX-4T-1 vector (Clontech, USA, California). Flag-tagged Prmt1 and Dr-VSP-GFP vectors were described in the previous reports (*Malkki, 2014*; *Keum et al., 2016*).

## Generation of *Prmt1* knockdown stable cell line

Stable cell lines exhibiting knockdown of Prmt1 were generated using Mission shRNA vector (Sigma, USA, Missouri). The PLK0.1 TR control vector (random 18mer) was used for generating the control cell lines. 293T cells were seeded at 3 × 10 (*Rogawski, 2000*) cells per 10 cm plate and transfected with 10 μg of the shRNA vector together with 5 μg of VSVg and 5 μg of Δ8.2 helper plasmids using Effectene (Qiagen, Germany, Hilden). The medium was changed 12 hrs after transfection. After 48 hrs, the lentivirus-containing culture media were harvested. 293T cells were infected by 0.45 μm-filtered supernatant from virus-producing cells in the presence of 8 μg/ml polybrene. After 2 days, the puromycin-resistant cells were maintained with 2 μg/ml puromycin. Medium containing puromycin was replaced every 2 days until puromycin-resistant stable cell lines were established.

## Cell culture, transfection and recording

Human embryonic kidney cell line HEK293T cells (ATCC, USA, Virginia) were maintained in Dulbecco's modification of Eagle's medium (Gibco-BRL, USA, California) supplemented with 10% fetal bovine serum at 37°C in 5% $CO_2$. For transient transfections, cells were transfected using Lipofectamine 2000 reagents (Invitrogen, USA, California) and green fluorescent protein was used as a reporter to label the transfectants. The KCNQ currents from HEK293T cells were measured with the whole-cell patch clamp technique. Voltage clamp was performed using an EPC-10 amplifier (HEKA Instruments, Germany, Lambrecht/Pfalz) at a sampling rate of 10 kHz filteredat 1kHz. Data were acquired using an IBM-compatible computer running Patchmaster software (HEKA Instruments, Germany, Lambrecht/Pfalz). The patch pipettes were pulled from borosilicate capillaries(Hilgenberg-GmbH, Germany, Malsfeld) using a Narishige puller (PC-10, Narishige, Japan, Tokyo). The patch pipettes had a resistance of 2–3 MW when filled with the pipette solution containing (in mM) 140 KMeSO$_4$, 20 KCl, 20 HEPES, 0.5 Na-GTP, 5 Mg-ATP, 4 vitamin C, and 10 1,2-bis (2-aminophenoxy) ethane *N,N,N_,N_*-tetraacetic acid (BAPTA), pH 7.4 adjusted with KOH. The normal external solution was as follows (in mM): 143 NaCl, 5.4 KCl, 5 HEPES, 0.5 NaH$_2$PO$_4$, 11.1 glucose, 0.5 MgCl$_2$, and 1.8 CaCl$_2$, pH 7.4 adjusted with NaOH. Pipette capacitance was compensated after formation of a gigaohm seal. Access resistance was typically 2.8–3.2 MΩ. The perfusion system was a homemade 100-μl perfusion chamber through which solution flowed continuously at 5 ml/min. The currents from HEK293T cells were studied by holding the cell at −60 mV, and 1-s steps from −70 to 40 mV in 10-mV increments were applied, followed by 1-s pulses to −60 mV. All recordings were carried out at room temperature (RT).

Currents were analyzed and fitted using *Patch master* (HEKA Instruments, Germany, Lambrecht/Pfalz) and Origin (ver. 6.0, Microcal, USA, Massachusetts) software. All values are given as mean ± standard error. I/V relationship were obtained by plotting the outward current at the end of a 1-s test pulse as a function of the test potential.

## Immunoprecipitation and immunoblotting

Cells were lysed in NETN lysis buffer containing 1 mM phenylmethylsulfonyl fluoride, 1 mM sodium fluoride, 1 mM sodium orthovanadate, 2 µg/ml aprotinin, 2 µg/ml leupeptin, and 1 µg/ml pepstatin A for 1 hr at 4°C. Following lysis, the cells were centrifuged at 13,000 rpm for 10 min at 4°C. Total cell extracts were incubated with appropriate antibodies for 4 hrs at 4°C and Protein G plus/Protein A-agarose (Calbiochem, USA, California) was added, and the incubation was continued for 2 hrs. After incubation, immunoprecipitates were washed three times with NETN lysis buffer and subjected to sodium dodecyl sulfate-polyacrylamide gel electrophoresis (SDS-PAGE) and transferred to nitro-cellulose membranes. The membranes were blocked with TBS containing 5% skim milk and 0.1% Tween 20 for 30 min at RT. The membranes were incubated with appropriate primary antibody for 2 hr at RT. After washing with TBS containing 0.1% Tween 20 three times for 10 min, the membranes were incubated with horseradish peroxidase-conjugated secondary antibody for 1 hr at RT. After washing with TBS containing 0.1% Tween 20 three times for 10 min, proteins were detected using the enhanced chemiluminescent (ECL) system (GE Healthcare, USA, Chicago). Primary antibodies used in this study were as follows: Prmt1 (1:1000, Millipore, USA, Massachusetts), ADMA (1:1000, Cell Signaling, USA, Massachusetts), α-tubulin (1:2000, Cell signaling, USA, Massachusetts), KCNQ2 (1:1000 Abcam, UK, Cambridge), Hsp90 and PRMT8 (1:1000, Santa Cruz, USA, Texas), α-Flag, α-Myc, α-HA and α-GST (1:1000, Abfrontier, South Korea, Seoul), and GAPDH (1:2000, Abfrontier, South Korea, Seoul).

## Peptide analysis with mass spectrometry

For mass analysis of KCNQ2 proteins, HA-KCNQ2 expression vectors were tranfected into HEK293T cells. Cells were lysed in NETN lysis buffer (50 mM Tris–HCl, pH 7.5, 150 mM NaCl, 0.5% Nonidet P-40) containing 1 mM phenylmethylsulfonyl fluoride, 1 mM sodium fluoride, 1 mM sodium orthova-nadate and 2 µg/ml aprotinin, 2 µg/ml leupeptin and 1 µg/ml pepstatin A for 1 hr at 4°C. Following lysis, cells were centrifuged at 13 000 r.p.m. for 10 min at 4°C. Total cell extracts were incubated with anti-HA antibody for 2 hrs at 4°C and protein G plus/protein A-agarose (Calbiochem, USA, California) was added, and the incubation was continued for 2 hrs. After incubation, immunoprecipitates were washed three times with NETN lysis buffer and subjected to SDS-PAGE electrophoresis. In-gel digestion was carried out with 12.5 ng/µl sequencing grade modified trypsin (Promega, USA, Wisconsin) in 50 mM $NH_4HCO_3$ buffer (pH 7.8) at 37°C for overnight. Produced tryptic peptides were extracted with 5% formic acid in 50% ACN solution at room temperature for 20 min. The supernatants were collected and dried with SpeedVac. Resuspended samples in 0.1% formic acid were purified and concentrated using C18 ZipTips (Millipore, USA, Massachusetts) before MS analysis.

The tryptic peptides were loaded onto a fused silica microcapillary column (12 cm × 75 µm) packed with C18 reversed phase resin (5 µm, 200 Å). Peptide separation was conducted with a series of step gradients composed of initial isobaric flow for 5 min with 3% solvent B (0.1% formic acid in acetonitrile), then linear gradient from 3% to 40% for 40 min. At the end of each running, 90% of solvent B was eluted for 10 min with the flow rate 250 nL/min. The% gradient of solvent B was against solvent A (0.1% formic acid in $H_2O$). The column was directly connected to LTQ linear ion-trap mass spectrometer (ThermoFinnigan, USA, New Jersey) equipped with a nano-electrospray ion source. The electrospray voltage was set at 1.95 kV, and the threshold for switching from MS to MS/MS was 500. The normalized collision energy for MS/MS was 35% of main radio frequency amplitude (RF) and the duration of activation was 30 ms. All spectra were acquired in data-dependent scan mode. Each full MS scan was followed by five MS/MS scan corresponding from the most intense to the fifth intense peaks of full MS scan. The acquired LC-ESI-MS/MS fragment spectra were searched against a modified NCBI protein reference database using Mascot program. The searching conditions were trypsin enzyme specificity, a permissible level for three missed cleavages, peptide tolerance; the amu, a mass error of amu on fragment ions, fixed modification of carbamidomethylation of cysteine, and variable modifications of oxidation of methionine, monomethylation of arginine and dimethyla-tion of arginine.

## In vitro methylation assay

Purified recombinant GST-KCNQ2 (amino acids 320–449) proteins were incubated for 3 hrs at 37°C with 1 µg of recombinant GST-Prmts in 30 µl methylation buffer (50 mM Tris/HCl, pH 7.5)

supplemented with 0.5 µCi of S-adenosyl-L-[methyl-[3]H]methionine (PerkinElmer Life Sciences, USA, Massachusetts). Reactions were stopped by adding 2× SDS-PAGE sample buffer and heating. Samples were analyzed by SDS-PAGE and fluorography. GST-KCNQ2 and GST-tagged Prmts were bacterially expressed purified. Myc-tagged Prmt5 was expressed in HEK293T cells and purified by immunoprecipitation and elution.

### Protein lipid overlay assay

Protein lipid overlay assays were performed as described previously (Dowler et al., 2002). Briefly, extracts were prepared from HEK293T control cells or Prmt1 knockdown cells transfected with HA-KCNQ2 expression vectors by resuspending in modified NETN buffer (20 mM Tris, pH 7.5, 150 mM NaCl, 0.5% Nonidet P-40, 1 mM EDTA, 1 mM sodium fluoride, 1 mM $Na_3VO_4$, 1 mM phenylmethyl-sulfonyl fluoride, 1 µg/ml Leupeptin, 0.2 µg/ml Pepstatin A). $PIP_2$ (10–500 pmol) was spotted on Hybond-C extra membrane (Amersham Biosciences, USA, New Jersey). Spotted membranes were blocked with 0.2% fatty acid-free BSA (Sigma USA, Missouri) in TBST for 1 hr at room temperature. The membranes were incubated with cell lysates for 2 hrs at room temperature, followed by washing and detection with anti-HA antibody.

### Statistical analysis

Data are presented as mean ± standard error of mean. Student's t-test or ANOVA Tukey tests were performed with Sigmaplot 12.0 after normal distribution of data was examined Shapiro-Wilk test. To detect a difference in mean value between two independent groups with 80% power, a sample-size of 4 cells in each group is needed when Student's t-test is used with parameter values that are typical for the present study (alpha = 0.05, the effect size d = 3). When ANOVA Tukey tests is used with parameter values that are typical for the present study (alpha = 0.05, the effect size f = 30), sample-size of 3 cells in each group is needed for a difference in mean value between two independent groups with 80% power. We used G*power program (Faul et al., 2009) (http://www.gpower.hhu.de/) for this estimation. We obtained data from more than 4 cells for Student's t-test or more than 3 cells for ANOVA Tukey test when statistical significance was tested.

## Acknowledgements

Authors thank Drs Mark S Shapiro and Byung-Chang Suh for kindly providing valuable reagents. This research was supported by Basic Science Research Program through the National Research Foundation of Korea (NRF) funded by the Ministry of Science, ICT & Future Planning and Technology (NRF-2012R1A2A2A01046878, NRF-2015R1A2A1A15051998 and 2015-048055).

## Additional information

### Funding

| Funder | Grant reference number | Author |
|---|---|---|
| National Research Foundation of Korea | NRF-2012R1A2A2A01046878 | Hyun-Ji Kim<br>Seul-Yi Lee<br>Hanna Kim<br>Jewoo Koh<br>Hana Cho |
| National Research Foundation of Korea | NRF-2015R1A2A1A15051998 | Myong-Ho Jeong<br>Tuan Anh Vuong<br>Jong-Sun Kang |
| National Research Foundation of Korea | 2015-048055 | Kyung-Ran Kim<br>Won-Kyung Ho |

The funders had no role in study design, data collection and interpretation, or the decision to submit the work for publication.

## Author contributions

H-JK, M-HJ, K-RK, C-YJ, Drafting and revising the article, Acquisition of data, Analysis and interpretation of data; S-YL, HK, JK, TAV, SJ, HY, S-KP, DC, KJK, J-WS, JMP, DJ, S-HK, W-KH, Conception and design, Acquisition of data, Analysis and interpretation of data; SHK, Acquisition of data, Analysis and interpretation of data; J-SK, S-TK, HC, Conception and design, Analysis and interpretation of data, Drafting or revising the article

## Author ORCIDs

Jewoo Koh, http://orcid.org/0000-0003-4977-3728
KyeongJin Kang, http://orcid.org/0000-0003-0446-469X
Won-Kyung Ho, http://orcid.org/0000-0003-1568-1710
Hana Cho, http://orcid.org/0000-0002-9394-8671

## Ethics

Animal experimentation: All animal experiments were approved by the Institutional Animal Care and Research Advisory Committee at Sungkyunkwan University School of Medicine Laboratory Animal Research Center (Approval No. IACUC-11-39).

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
