## [Decision Letter]

Thank you for submitting your article "Protein Arginine Methylation Facilitates KCNQ Channel-PIP_2_ Interaction Leading to Seizure Suppression" for consideration by *eLife*. Your article has been reviewed by two peer reviewers, and the evaluation has been overseen by a Reviewing Editor and Richard Aldrich as the Senior Editor. One of the experts involved in the review of your submission has agreed to reveal his identity: Gary Yellen.

The reviewers have discussed the reviews with one another and the Reviewing Editor has drafted this decision to help you prepare a revised submission.

Summary:

This study demonstrates the consequences and necessity of protein arginine methylation by PRMT1 on KCNQ2/3 (M-current) channels, mutations of which have previously been linked to epilepsy, and on seizures in mice heterozygous for PRMT1. The PRMT1 +/- mice are shown to have spontaneous seizures as well as a change in excitability of dentate granule neurons of the hippocampus, accompanied by a smaller M-current in those neurons. Through a series of electrophysiological and biochemical evidence, the authors show that KCNQ2 channels are indeed methylated and directly interact with PRMT1. The targeted arginines are within a previously identified KCNQ2 PIP_2_ binding site, leading to the proposal that methylation of KCNQ2 channels alters their affinity for PIP_2_. likely producing changes in their probability of opening.

Essential revisions:

Both reviewers found the main findings of the manuscript to be interesting and convincing. Certain parts of the study, however, require further clarification and/or experimentation. The key points that must be addressed are as follows:

1) The evidence presented makes it seem likely that PRMT1 may influence more channels than just KCNQ. Please consider and acknowledge the possibility and/or likelihood of PRMT1 actions at other K channels, especially taking into account the quantitative elements of the changes observed (detailed below in Reviewers' Specific Point 1).

2) The interpretation of the PIP_2_ affinity experiments raised a number of questions. Please improve the measurements of PIP_2_ affinity, ideally by new experiments with CiVSP and/or by clarifying the apparent quantitative inconsistencies within the dataset (in Figure 6 and Figure 7) and differences from the literature (detailed below in Reviewers' Specific Point 2).

3) Aspects of the results with peroxide were unclear. Please clarify by discussing why methylation appears required for the peroxide-mediated augmentation of current, even though the effect of methylation can occur in channel mutants not susceptible to peroxide modulation (detailed below in Reviewers' Specific Point 3).

The Reviewers' General Comments are also included for your information.

Reviewers' General Comments:

This well-designed study shows a dramatic and interesting effect of protein arginine methylation on the epilepsy in mice and on the KCNQ2/3 (M-current) channels, mutations of which have previously been linked to epilepsy. The PRMT1 +/- mice are shown to have spontaneous epileptic seizures as well as a dramatic change in the excitability of the dentate granule neurons of the hippocampus. The authors nicely show that PRMT1 associates with the KCNQ2 channel, or with its isolated C-terminus, by immunoprecipitation. They use LC-MS to identify the four arginines that are methylated in KCNQ2, and to show that substituting 2 to 4 of these arginines reduces (or eliminates) the methylation. These mutations also dramatically reduce the KCNQ2 currents in a heterologous system, though they did not eliminate protein expression or the current in the presence of the channel activator retigabine.

The goal of this study is to understand the mechanism by which loss of PRMT1, an arginine methyltransferase, leads to hyperexcitability and seizures in mice. In particular, the authors show that dentate granule cells from PRMT1 heterozygous mice fire a greater number of action potentials and have increased input resistance. Based on this data, they speculated loss of PRMT1 most likely caused a decreased to the M-current (KCNQ2/3), a potassium conductance critical for neuronal excitability. This is the first demonstration of KCNQ2 channels being targeted for methylation. The evidence that KCNQ2 channels are methylated is convincing.

Reviewers' Specific Points (expanding on essential revisions):

1) While the conclusions about PRMT1 effects on KCNQ2 are convincing, there is clearly another major contribution to the altered DGN excitability. This is not denied by the authors, but it is practically ignored until the penultimate sentence of the paper. It would be much better to acknowledge this contribution much earlier in the paper, and then to have a slightly more complete discussion of the differences between the PRMT1 +/- phenotype, the only partial ability to mimic it with XE991 blockade of KCNQ2, and the only partial ability to reverse it with PIP_2_.

In fact, the authors show that PRMT1 deficiency shifts the firing frequency of DGC from 8Hz to 31Hz, an almost 4 fold change (Figure 2). They attribute this change to the complete absence of M-current in PRMT1+/- DGC. However, application of XE991 at a concentration that completely blocks the M-current (Figure 3) only shifts the AP frequency in wild-type cells from 8Hz to 13Hz. As such, what evidence do the authors have that no additional potassium currents that are also PIP_2_ dependent (SK, sAHP) and are known to control DGC firing properties did not change in cells deficient in PRMT1?

Further, the authors argue that the M-current is likely the main culprit for the change in the firing rate in PRMT1 het mice because application of XE991 in PRMT1+/- DGC did not have any additional effect. However, the lack of change in the firing rate by XE991 might be due to a ceiling effect as the PRMT1+/- DGC fired at 34Hz when XE991 was applied, unlike the control cells that fired at 8Hz. The authors should have tested the effect of XE991 in neurons that fired the same number of action potentials by simply adjusting the injected current. The same ceiling issue is for all experiments done with XE991 in the presence of different PRMT1 blockers (Figure 4). By not controlling the number of AP, one can get very misleading results.

2) To correctly quantify the change in PIP_2_ affinity in KCNQ2 mutants (R to K substitutions), the authors should have co-transfected their cells with the KCNQ2 mutant channels and Ci-VSP, a voltage-activated PIP_2_ phosphatase (currently, the standard methodology). Using this approach the rate and extent of KCNQ2/3 current inhibition following PIP_2_ depletion as well as the rate of current recovery following PIP_2_ restoration could be measured. Also relating to this point, the authors state that inclusion of 200uM diC8-PIP_2_ in the pipette did not increase KCNQ2 currents suggesting that endogenous PIP_2_ might be at saturating concentrations (subsection Arginine methylation of KCNQ2 is required for maintaining PIP_2_ affinity”, third paragraph). This finding is inconsistent with all previous papers on KCNQ2 channels and PIP_2_. See work by Shapiro and colleagues on PIP_2_ as well as Brown DA and colleagues on PIP_2_ and KCNQ2 and KCNQ2/3 channels. All papers agree that increasing endogenous PIP_2_ levels leads to an increase in KCNQ2/3 currents, as their PIP_2_ affinity is low.

3) Regarding the experiments with hydrogen peroxide (Figure 9), it is known that the hydrogen peroxide induced increase in KCNQ2/3 currents is due to its action on a series of cysteines in the KCNQ2 and not through arginines on the S6 (Gamper et al.. 2006), cited by the authors. So, it is unclear why mutating S6 arginines alone leads to abrogation of the hydrogen peroxide effect on KCNQ2 channels (Figure 9). Please clarify.

---

## [Author Response]

*Reviewers' Specific Points:*

*1) While the conclusions about PRMT1 effects on KCNQ2 are convincing, there is clearly another major contribution to the altered DGN excitability. This is not denied by the authors, but it is practically ignored until the penultimate sentence of the paper. It would be much better to acknowledge this contribution much earlier in the paper, and then to have a slightly more complete discussion of the differences between the PRMT1 +/- phenotype, the only partial ability to mimic it with XE991 blockade of KCNQ2, and the only partial ability to reverse it with PIP_2_.*

As the reviewer pointed out, XE991 treatment did not fully mimic the firing rates of PRMT1+/- GCs although it significantly increased firing rates over the entire range of current injection in WT GCs. In addition to firing rates, we analyzed all the parameters of intrinsic firing properties including threshold current, input resistance, and resting membrane potential. PRMT1+/- GCs required ~90pA less current injection than WT GCs to achieve AP, which we think facilitates their AP initiation and, hence, increase in spike number. The change in the threshold currentin PRMT1+/- neurons reflects the change of input resistance. Thus, it is of note that, even though XE991 treatment fully recapitulates firing rates observed in PRMT1+/- mice, it does increase input resistance to the level that was not different from that in PRMT1+/- mice. On the other hand, XE991 reduced threshold current by ~30pA, which is less effective than PRMT1 blocker.

DiC8-PIP_2_ loading *reversed firing rates* in PRMT1+/- GCs (<0.05). For example, with 200pA current injection for 1 sec, WT GCs fired 9 APs while PRMT1+/- GCs fired 31 APs, which returned to 15 APs with diC8-PIP_2_ loading. To analyze PIP_2_ effects in detail, we examined threshold current and input resistance. With diC8-PIP_2_, threshold current was also reversed from 115 pA to 175pA, which is close to the WT level, 199pA. Thus firing rates and threshold current are rescued significantly but incompletely by diC8-PIP_2_. Input resistance, however, was fully rescued to the WT level by PIP_2_ loading (p>0.05 vs WT, Figure 7).

Thus, there might be another factor(s) contributing to reduction of threshold current and increase of firing rates. We suspected the persistent Na^+^ current or Na^+^ leak channel because their activation might afford the additional reduction of threshold current with little effect on input resistance. Regardless of this possible explanation, our study showed the solid link between PRMT1 and KCNQ/M channels taking advantage of various approaches involving in situ and heterologous electrophysiology.

As the reviewers suggested, we acknowledge the possible involvement of channels other than KCNQ in the Results section of the revised manuscript and have added a slightly more complete discussion about this point in the Discussion section.

In the Results:

“Thus, these results showed that PRMT1+/- GCs displayed a high firing rate at baseline, and their firings did not further increase during XE-991 application, suggesting that defective M-current contributes to the neuronal hyperexcitability observed in the PRMT1+/- mice. However, we cannot exclude the potential involvement of other PRMT1 target(s) in the neuronal hyperexcitability observed in the PRMT1+/- mice.”

**“**Although the recovery of excitability with PIP_2_ application was significant it was incomplete, suggesting the possible involvement of other PRMT1 target(s) in neuronal hyperexcitability. […] Further study will be required to identify additional targets of PRMT1.”

In the Discussion:

**“**Consistent with this, our results show that the effects of KCNQ channel block on firing rates are not the same as those of PRMT1 block (Figure 3), implying a possible involvement of other PRMT1 target(s) in neuronal hyperexcitability and seizures. PIP_2_ loading to PRMT1+/- GCs recovered the firing properties significantly but incompletely. It appears that SK and BK channels are not involved in this matter. Thus further studies are required to identify other PRMT1 targets in the control of neuronal excitability.”

In fact, the authors show that PRMT1 deficiency shifts the firing frequency of DGC from 8Hz to 31Hz, an almost 4 fold change (Figure 2). They attribute this change to the complete absence of M-current in PRMT1+/- DGC. However, application of XE991 at a concentration that completely blocks the M-current (Figure 3) only shifts the AP frequency in wild-type cells from 8Hz to 13Hz. As such, what evidence do the authors have that no additional potassium currents that are also PIP_2_ dependent (SK, sAHP) and are known to control DGC firing properties did not change in cells deficient in PRMT1?

SK and BK channels are known to regulate membrane excitability of dentate gyrus GCs (Nat Neurosci. 8, 1752-1759) and are also PIP_2_-dependent (Nat Chem Biol. 10, 753-759; J Biol Chem. 289, 18860-18872). Using apamin and paxilline, specific SK and BK channel blockers, respectively, we examined whether SK and/or BK channels were altered in PRMT1-deficient GCs. Consistent with previous studies (Nat Neurosci. 8, 1752-1759), the AP firing rate of control WT GCs was markedly increased by SK channel blockade (see attached figure below) whereas paxilline had little effect on control WT GCs. Similar responses were observed in dentate gyrus GCs pretreated with furamidine, a PRMT1 inhibitor. They displayed a high firing rate at baseline, and their firings further increased by apamin application, but not by paxilline. These data illustrate that neither SK nor BK channel activities are altered in PRMT1-deficient GCs.

We added new data as Figure 7—figure supplement 5, and included the following in the Results:

**“**As immediate candidates, we tested other K^+^ channels such as SK and BK channels in PRMT1 effects which are known to regulate membrane excitability of dentate gyrus GCs (Brenner et al., 2005). The data obtained by using apamin and paxilline, specific SK and BK channel blockers, respectively, ruled out that the contribution of these channels in hyperexcitability of PRMT1-deficient GCs (Figure 7—figure supplement 5).”

Further, the authors argue that the M-current is likely the main culprit for the change in the firing rate in PRMT1 het mice because application of XE991 in PRMT1+/- DGC did not have any additional effect. However, the lack of change in the firing rate by XE991 might be due to a ceiling effect as the PRMT1+/- DGC fired at 34Hz when XE991 was applied, unlike the control cells that fired at 8Hz. The authors should have tested the effect of XE991 in neurons that fired the same number of action potentials by simply adjusting the injected current. The same ceiling issue is for all experiments done with XE991 in the presence of different PRMT1 blockers (Figure 4). By not controlling the number of AP, one can get very misleading results.

In each condition, spike trains in dentate gyrus granule cells were evoked by injecting 1s depolarizing current pulses of different intensities (from 100 to 400pA, 50-pA increment) into the cell before and after application of XE991. Figure 3 show the mean number of action potentials (AP No.) plotted against the wide range of eliciting currents (100-400 pA) although we presented only the mean AP No. against 200pA in text. As you may see in Figure 3, at 100pA current injection, the mean AP firing frequency in PRMT1+/- GCs was 8Hz, which is the same number of APs in WT GCs at 200-pA current injection. In contrast to WT GCs, however, it did not further increase upon XE991 treatment (P>0.05, Figure 3). The same applies to GCs treated with different PRMT1 blockers. In Figure 4, at 100-pA current injection, MTA- or furamidine-treated GCs fired at 10Hz and 6HZ, respectively and they did not respond to XE991 treatment (P>0.05). These data suggest that the lack of change in the firing rate by XE991 in methylation-deficient GCs is not due to a ceiling effect, but rather due to loss of M currents. In addition, the increase of firing rates in furamidine-pretreated GCs in response to apamin also rules out the ceiling effects (see point #1-2).

*2) To correctly quantify the change in PIP_2_ affinity in KCNQ2 mutants (R to K substitutions), the authors should have co-transfected their cells with the KCNQ2 mutant +channels and Ci-VSP, a voltage-activated PIP_2_ phosphatase (currently, the standard methodology). Using this approach the rate and extent of KCNQ2/3 current inhibition following PIP_2_ depletion as well as the rate of current recovery following PIP_2_ restoration could be measured.*

As suggested, we co-transfected HEK293T cells with KCNQ2 channels and voltage-activated phosphatase (Dr-VSP, a zebrafish Ci-VSP homolog). Both Ci-VSP and Dr-VSP were widely used for quantification of PIP_2_ affinity (Front Pharmacol 2015 Jun 19;6:127). Consistent with previous reports (J Gen Physiol. 135, 99-114), WT KCNQ2 current was reduced by activation of VSP and fully recovered after repolarization (see Figure 7—figure supplement 2). The recovery of WT KCNQ2 current was fitted with a single exponential; Mean recovery time constants for current were 7.0 ± 1.1s (n=9). When subjecting RK mutants to the same VSP activation protocol, the overall behavior appeared to be similar to the WT KCNQ2 channel. The remaining current of RK mutants was small but still responsive to VSP activation. However, recovery of RK mutants was slower than that of WT channels and it was not well fitted with the single-exponential function. Thus, to compare recovery kinetics between WT KCNQ2 channels and RK mutants, we calculated the time to the 70% maximum current (T_70_). Compared to T_70_ of WT KCNQ2 channels, T_70_s of RK mutants were significantly increased (P<0.05). Taken together, these data are in alignment with the conclusion that hypomethylation of KCNQ2 channels diminishes PIP_2_ affinity of KCNQ channels.

We added new data as Figure 7—figure supplement 2, and modified the text in the Results to read:

“Altered PIP_2_ affinity in RK mutants was also assessed by a voltage-sensitive phosphatase from *Danio rerio* (Dr-VSP), which hydrolyzes PIP_2_ at highly depolarized voltages (e.g., +100 mV) and transiently reduces the PIP_2_ level (Falkenburger et al., 2010; Rjasanow et al., 2015). Dr-VSP was coexpressed with WT KCNQ2 or RK mutants, and its activity was elicited by membrane depolarization. Consistent with a previous report (Falkenburger et al., 2010), activation of Dr-VSP reduced WT KCNQ2 currents, which were recovered quickly after PIP_2_ resynthesis on repolarization (Figure 7—figure supplement 2). When subjecting RK mutants to the same VSP activation protocol, the overall behavior was similar to the WT KCNQ2 channel; however, recovery of current was slowed 2-4 folds (Figure 7—figure supplement 2). The mean time to the 70% maximum current (T_70_) for R333K, R345K, R353K and R435Kwas 22.8 ± 3.7, 24.1 ± 1.9, 32.1 ± 2.6, and 17.9 ± 1.5 s, respectively (P<0.05 vs WT:9.5 ± 3.1 s). The slowed recovery after VSP reflects the reduced PIP_2_ affinity (Rjasanow et al., 2015), further supporting for the reduced PIP_2_ affinity of RK mutants.”

*Also relating to this point, the authors state that inclusion of 200uM diC8-PIP_2_ in the pipette did not increase KCNQ2 currents suggesting that endogenous PIP_2_ might be at saturating concentrations (subsection Arginine methylation of KCNQ2 is required for maintaining PIP_2_ affinity”, third paragraph). This finding is inconsistent with all previous papers on KCNQ2 channels and PIP_2_. See work by Shapiro and colleagues on PIP_2_ as well as Brown DA and colleagues on PIP_2_ and KCNQ2 and KCNQ2/3 channels. All papers agree that increasing endogenous PIP_2_ levels leads to an increase in KCNQ2/3 currents, as their PIP_2_ affinity is low.*

In fact, Brown DA and colleagues reported that 200μM diC8-PIP_2_ produced a significant reduction in the inhibitory effect of oxotremorine-M on M-currents. However diC8-PIP_2_ at 200μM did not affect the M-current density itself (J Neurosci. 26, 7950-7961) which is consistent with our data showing that whole-cell dialysis of 200μM diC8-PIP_2_ does not increase M currents in DG granule cells.

The discrepancy between the work from Shapiro laboratory and our current work might be due to the experimental designs used. Work from the Shapiro laboratory inquired about EC50 for single channel in CHO cell-excised inside-out patches (J Neurosci. 25, 9825-9835), while we examined the effects of diC8-PIP_2_ on whole cell KCNQ currents. As several cellular factors can modulate PIP_2_ sensitivity, EC50 value for single channel recording in inside-out patches can be different from that for whole-cell currents. Another possibility is the variable saturating concentration of PIP_2_ for KCNQ channels in different cell systems. For example, EC50 value for KCNQ2/KCNQ3 channels in CHO cells-excised patches was 40 μM (J Neurosci. 25, 9825-9835) while it was 87mM in oocytes-excised macropatches (Neuron 6, 963-975).

3) Regarding the experiments with hydrogen peroxide (Figure 9), it is known that the hydrogen peroxide induced increase in KCNQ2/3 currents is due to its action on a series of cysteines in the KCNQ2 and not through arginines on the S6 (Gamper et al. 2006), cited by the authors. So, it is unclear why mutating S6 arginines alone leads to abrogation of the hydrogen peroxide effect on KCNQ2 channels (Figure 9). Please clarify.

As the reviewers point out, triple cysteines are required for augmentation of KCNQ channel activity induced by H_2_O_2._ Given that mutation of S6 arginines in KCNQ2 abrogates its PIP_2_ binding, we propose an additional regulatory mechanism involving PIP_2_ interaction of KCNQ2 channel for H_2_O_2_ effects.